# Gold Nanoparticles for Retinal Molecular Optical Imaging

**DOI:** 10.3390/ijms25179315

**Published:** 2024-08-28

**Authors:** Sumin Park, Van Phuc Nguyen, Xueding Wang, Yannis M. Paulus

**Affiliations:** 1Department of Biomedical Engineering, University of Michigan, Ann Arbor, MI 48105, USA; suminp@umich.edu; 2Department of Ophthalmology and Visual Sciences, University of Michigan, Ann Arbor, MI 48105, USA; vnguye69@jh.edu; 3Department of Ophthalmology, Johns Hopkins University, Baltimore, MD 21287, USA; 4Department of Biomedical Engineering, Johns Hopkins University, Baltimore, MD 21287, USA

**Keywords:** molecular imaging, gold nanoparticles, retinal imaging, optical imaging, photoacoustic microscopy, optical coherence tomography, fluorescence imaging

## Abstract

The incorporation of gold nanoparticles (GNPs) into retinal imaging signifies a notable advancement in ophthalmology, offering improved accuracy in diagnosis and patient outcomes. This review explores the synthesis and unique properties of GNPs, highlighting their adjustable surface plasmon resonance, biocompatibility, and excellent optical absorption and scattering abilities. These features make GNPs advantageous contrast agents, enhancing the precision and quality of various imaging modalities, including photoacoustic imaging, optical coherence tomography, and fluorescence imaging. This paper analyzes the unique properties and corresponding mechanisms based on the morphological features of GNPs, highlighting the potential of GNPs in retinal disease diagnosis and management. Given the limitations currently encountered in clinical applications of GNPs, the approaches and strategies to overcome these limitations are also discussed. These findings suggest that the properties and efficacy of GNPs have innovative applications in retinal disease imaging.

## 1. Introduction

Gold nanoparticles (GNPs) have been identified as a promising contrast agent in retinal imaging among recent innovations [1,2]. Their unique optical properties, including tunable surface plasmon resonance, size, shape-dependent optical absorption, and scattering, enhance the contrast and clarity of imaging modalities [3,4]. These properties make GNPs well suited for retinal imaging, improving the resolution of existing imaging and allowing for more detailed visualization of complex retinal structures and functions, providing a better understanding of the mechanisms involved in retinal diseases [5,6,7,8]. Gold has displayed a wide range of properties throughout history and has proven to be highly suitable for biomedical applications due to its unique characteristics. GNPs demonstrate unique attributes such as biocompatibility, low toxicity, strong functionalization ability, improved surface plasmon resonance, and optical absorption and scattering, as well as localized thermal effects. These characteristics render GNPs exceptionally well suited for various applications, ranging from diagnostic imaging to therapeutic interventions. GNPs exhibit a significant level of transparency in biological tissue, thereby enhancing detection capabilities across different imaging modalities. Numerous synthesis methods and surface modification materials are available for achieving these features, which can vary based on the intended applications. As a result, the characteristics of GNPs can be modified accordingly. Due to their unique properties and capabilities, GNPs have attracted attention and have been studied in ophthalmology for imaging enhancement and disease treatment.

Recent advancements in imaging technology have significantly transformed the field of ophthalmology, particularly in the diagnosis and monitoring of retinal diseases [9,10]. High-resolution and non-invasive technologies like photoacoustic microscopy (PAM), optical coherence tomography (OCT), OCT angiography (OCTA), and fluorescence imaging (FI) have garnered increased interest as fundamental technologies. They offer a comprehensive view of retinal structures and aid in monitoring disease progression and treatment response. PAM plays a crucial role in comprehending diseases at a microenvironmental level by visualizing deep ocular structures (e.g., size and shape, texture, and pattern) and providing functional information (e.g., blood flow, oxygen saturation, oxygenated and deoxygenated hemoglobin, oxygen metabolism) [11,12,13,14,15,16]. PAM utilizes short laser pulses to trigger thermoelastic expansion in tissues, thereby generating ultrasound waves that facilitate the creation of precise images [17,18]. OCT provides valuable insights into intricate retinal structures such as retinal layers, optic nerve morphology, and the presence of fluid or lesions, while OCTA offers a comprehensive perspective by providing details on both structural and functional aspects, including blood flow, vascular network mapping, and perfusion dynamics [19,20,21,22,23,24]. FI, which includes fundus autofluorescence and fluorescein angiography, enhances diagnostic capabilities by allowing visualization of the retinal and choroidal circulations, metabolic changes, and structural abnormalities of the eye. This technology offers data on the retina and microvasculature, pinpointing areas of leakage or non-perfusion, and identifying abnormal accumulations of fluorescent substances within the eye [25,26,27]. These imaging techniques improve the timely identification, precise diagnosis, and successful treatment of retinal diseases, such as age-related macular degeneration (AMD), retinal neovascularization (RNV), choroidal neovascularization (CNV), diabetic retinopathy, and retinal vein occlusion. They aid in tailoring treatment strategies to individual patients, as well as monitoring the progression of the disease and response to treatment [28,29,30]. Therefore, these improvements highlight the important role of imaging in enhancing patient outcomes by increasing diagnostic precision, customizing treatments to individual cases, and raising the overall standard of care.

Despite their potential, the integration of advanced optical imaging techniques and GNPs into retinal imaging faces constraints. Limitations such as restricted penetration depth, resolution, and the ability to differentiate tissues or molecules with high contrast in pathology have hindered the early detection and precise diagnosis of retinal diseases. Researchers continue the development and tailoring of GNPs to overcome limitations by improving imaging contrast, resolution, size control, surface modifications, and other modifiable factors. The potential of GNPs in optical retinal imaging stems from their ability to enhance image quality and contrast, as well as their adaptability for targeted molecular imaging [1,31]. This advancement of GNPs is anticipated to contribute to the elucidation of the complex characteristics of retinal structures and pathologies, thereby facilitating a detailed understanding and treatment of retinal diseases. In conclusion, this aspect of GNPs presents novel opportunities in ophthalmology by enabling the accurate imaging of distinct retinal cells or layers and the prompt identification of molecular disease indicators.

This paper aims to explore the various functions of GNPs in retinal imaging, discussing their synthesis, properties as contrast agents, and applications to determine their efficacy (Figure 1). By delving into the mechanisms by which GNPs enhance optical imaging signals and qualities and evaluating their impact across various imaging modalities, this paper reviews the potential of GNPs in clinical applications and addresses the pertinent challenges that accompany the adoption of GNPs in clinical translation. This review intends to emphasize the potential of GNPs and their role in advancing the management of retinal diseases as a foundational element in the development of retinal imaging technologies.

## 2. Gold Nanoparticles (GNPs): Synthesis and Properties as Contrast Agents

### 2.1. Principles and Classification of GNPs

Gold nanoparticles (GNPs) in the size range of 1 to 100 nm have attracted great attention in the field of nanotechnology because of their notable optical, electronic, physicochemical, biological, and molecular characteristics [32]. These properties are intrinsically influenced by the dimensions, geometries (spheres, rods, stars, cubes, chains), and constituent materials, playing a critical role in determining the behavior of the particles for targeted applications [33,34]. The various morphologies of GNPs have been investigated and reported for several decades, including nanospheres, nanorods, nanoshells, nanocages, nanocubes, and nanoprisms (Figure 2) [35]. Furthermore, GNPs can be classified according to their synthetic method, size, and structural configurations, in addition to morphology [32,33,34]. This paper will categorize GNPs based on their morphology, focusing on how these shapes influence their optical characteristics and interactions with biological systems. Moreover, the structures of GNPs play a significant role in defining their strengths and drawbacks. Table 1 shows the representative classifications of GNPs based on the morphology and outlines the synthetic methods and their applications.

Moreover, the numerous synthetic methods allow for the formation of different morphologies in GNPs. Common methods include seed-mediated growth, citrate reduction, seedless, surfactant-free methods, thermal reduction, and laser ablation-based approaches [36,37,38,39]. The seed-mediated growth technique involves a two-stage process starting with the reduction of a gold salt solution to form seeds, followed by further growth using additional gold salt and a stabilizer like cetyltrimethylammonium bromide (CTAB) [40]. This method is proficient in producing anisotropic particles such as nanorods and nanostars through precise control of growth conditions. The citrate reduction method, a simple and widely used technique, reduces gold (III) chloride with sodium citrate, producing spherical GNPs with a consistent size distribution. Citrate serves as both a reducing agent and stabilizer, producing electrostatic stabilization to prevent aggregation [41]. The seedless method allows for simultaneous nucleation and growth of GNPs without pre-synthesized seeds, controlled by parameters such as gold salt concentration, reducing agent strength, and reaction temperature [42]. This method can generate nanostars and non-spherical GNPs, which are beneficial for applications necessitating improved surface area and optical characteristics. Surfactant-free methods focus on environmental safety and biocompatibility by excluding surfactants using temperature and pH regulation to ensure stability [43]. This simplifies purification and improves biocompatibility, making the GNPs suitable for in vivo applications. Thermal reduction involves heating a gold salt solution with a reducing agent, producing GNPs with high purity and uniformity. This method is scalable and suitable for applications requiring large quantities of GNPs [44]. Furthermore, laser ablation-based self-assembly uses high-energy laser pulses to vaporize a solid gold target in a liquid medium, resulting in the formation of nuclei of GNPs with enhanced condensation of gold atoms [7,45]. This approach provides an environmentally friendly option that ensures colloidal stability, ideal for sensitive biomedical applications. ijms-25-09315-t001_Table 1Table 1Classification of GNPs according to their morphology and various applications.MorphologySynthetic MethodsSize (nm)Maximum Absorption (nm)ApplicationsRef.NanosphereSeed-mediated growth method 30530Photothermal cancer treatment[46] 53560Residual fungicide detection[47]15520Photoacoustic imaging[48]26–50-X-ray CT and fluorescence imaging[49]10560Contrast agent for photoacoustic imaging[50]Citrate reduction method15520–526Antibiofilm nanomaterial[51]12, 18517, 522Lateral flow assay[52]NanorodSeed-mediated growth method-800Stem cell photoacoustic imaging[53]10 × 50,10 × 59 900, 980Cell labeling[54]51 ± 5 × 23 ± 3642Sonodynamic therapy[55]-818Contrast agent for ultrasound and photoacoustic imaging[56]35 ± 2 × 9 ± 2783pesticide thiram detection[57]NanostarSeed-mediated growth method200850Photoacoustic imaging-guided photothermal therapy[58]55 ± 5790Photoacoustic imaging[59]95 ± 7820Lateral flow immunoassay[60]126 ± 2.9544Photocatalytic Water Remediation[61]-520–650Colorimetric immunoassay[62]One-pot method45720Photoacoustic imaging[63]Surfactant-free growth method47 ± 17718Cellular imaging[64]58, 78600, 750Cancer diagnostic and therapeutic agent[65]NanoshellOldenburg method135.98 ± 2.36799Chemophotothermal theragnostic agent[66]Seed-mediated growth method-845Drug release monitoring[67]22539MR imaging and photothermal therapy[68]271 ± 19-Cancer diagnostics agent[69]Thermal reduction-523Exosome assay[70]Nanochain and nanoclusterpulsed laser ablation-based self-assembly64 × 20650PAM, OCT imaging contrast agent[7]30.0 ± 2.1 × 7.8 ± 1.1 610Imaging photosensitizer for PAM, OCT, fluorescence[8]NaOH-mediated NaBH_4_ reduction>5.5274, 398Imaging-guided cancer immunotherapy[71]Microwave-assisted method3.2 ± 0.4-Contrast agents for confocal fluorescence imaging[72]


### 2.2. Physicochemical and Optical Properties

The physicochemical characteristics of GNPs are influenced by their morphology, size, surface modifications, and synthesis method. These factors play a crucial role in determining the effectiveness of contrast agents, impacting their behavior in biological systems. The optical properties of GNPs, including absorption, scattering, and surface plasmon resonance (SPR), are significantly influenced by their size [73]. Smaller GNPs tend to absorb in the visible spectrum, while larger GNPs shift their absorption towards the near-infrared region (NIR region, 780 nm to 2500 nm), which can be advantageous for enhancing tissue penetration in biomedical imaging [74]. Surface modifications are also crucial in improving the durability and compatibility with biological systems of GNPs. Applying biologically active layers to GNPs results in both stabilization and the addition of functional groups essential for the targeting of specific cells or molecules. By leveraging these strategies, it is possible to finely tune the physicochemical and optical properties of GNPs, thereby optimizing their functionality in diverse fields such as medical diagnostics and therapeutic interventions. This section primarily focuses on the impact of morphology on the physicochemical and optical properties of GNPs. Different shapes, such as spheres, rods, stars, shells, chains, and clusters, possess distinct surface areas and geometric structures that affect their interactions with light and biological properties.

Spherical gold nanoparticles, known as the fundamental form of GNPs, have been extensively studied because of their symmetrical shape, leading to consistent SPR absorption in the visible region at various angles of incidence. The term SPR refers to the resonant oscillation of conduction electrons at the surface of the nanoparticles when exposed to incident light [75]. This consistency renders them well suited for applications necessitating constant optical characteristics, such as photothermal therapy, fungicide identification, and photoacoustic imaging. Nhat and colleagues developed gold nanospheres to be utilized as an SPR-based sensor for the real-time identification of remaining fungicides (Figure 3A) [47]. The gold nanospheres possess an average diameter of approximately 53 nm, exhibiting a distinct resonance peak at 560 nm. The synthesized gold nanospheres demonstrated more scattering light in larger sizes due to larger optical cross-sections and the ratio of scattering to total extinction. The gold nanospheres were coated with thiophanate methyl and the surface-enhanced Raman scattering (SERS) effect of them was observed clearly. Therefore, these gold nanospheres showed an increase in Raman signal intensity of the fungicide, indicating a promising approach for detecting residual fungicide in situ with heightened sensitivity. In addition, Kim et al. focused on improving optical properties through the use of gold nanospheres for identifying human matrix metalloproteinase-9 (hMMP-9), a protein associated with extracellular matrix modification, cancer cell invasion, and metastasis (Figure 3B) [48]. This study showcased the precise detection of MMP-9 via ultrasound-mediated photoacoustic imaging with gold nanospheres conjugated with a DNA aptamer. The aggregation of gold nanospheres in the presence of hMMP-9 enhances their optical absorption in the near-infrared window I, ranging from 680 to 970 nm. This improves the sensitivity and specificity of photoacoustic imaging for detecting tumors within their microenvironment. The photoacoustic spectrum of the contrast agent with hMMP-9 exhibited a 10-fold increase in a photoacoustic signal at 700 nm compared to bovine serum albumin or human matrix metalloproteinase-7 (hMMP-7). This study emphasizes the optical absorption properties of gold nanospheres, demonstrating their utility as contrast agents for photoacoustic imaging. However, nanospheres have limitations in their ability to localize and enhance electromagnetic fields compared to other shapes, attributable to their relatively simple geometry. Trakoolwilaiwan et al. utilized 12 nm diameter gold nanospheres for the development of a thermochromic lateral flow assay (LFA) tool, which was conjugated with temperature-sensitive dyes and antibodies that target the NS1 protein (Figure 3C) [52]. The thermal properties of various sizes and shapes of nanoparticles were assessed using a broad-spectrum LED, with the 12 nm gold nanospheres demonstrating the highest temperature elevation, attributed to their density and spherical morphology. This characteristic enhances the sensitivity of the rapid diagnostic LFA.

The rod-shaped GNPs exhibit a unique aspect ratio in comparison to nanospheres, with elongation along one dimension while maintaining similar dimensions in the other two axes. Gold nanorods possess distinctive optical properties as a result of their anisotropic shape, leading to the manifestation of two distinct SPR bands in the visible and NIR spectra. The longitudinal bands exhibit variation in accordance with the aspect ratio of nanorods, leading to distinct observable colors, thus offering advantages for a variety of biomedical applications. These characteristics make them suitable candidates for photothermal therapy and photoacoustic imaging, as well as diagnostic tools and contrast agents across a range of imaging techniques [33,34]. Salah et al. utilized modified gold nanorods to improve cell targeting and labeling through the seed-mediated growth technique, demonstrating significant advancement in stem cell photoacoustic imaging (Figure 4A) [53]. The synthesized gold nanorods show an extended LSPR peak at 800 nm. Thiolated-polyethylene glycol (PEG) functionalization was applied to the surface of gold nanorods, resulting in an increase in the particle size. This modification significantly enhanced the optical characteristics of gold nanorods, increasing their efficacy as a contrast agent. Due to their unique morphology, gold nanorods demonstrate longitudinal and transverse SPR phenomena arising from the oscillation of electrons along their respective long axis and short axes. The study described an improvement in signal intensity, as the altered gold nanorods demonstrated a notable enhancement in photoacoustic contrast. These findings highlight the potential application of modified gold nanorods as contrast agents, particularly in the context of innovative tracking imaging techniques in bio- and photoacoustic imaging. In addition, Jia and colleagues developed a contrast agent consisting of gold nanorods through seed-mediated growth methods for cellular labeling in Fourier-domain optical coherence tomography (Figure 4B) [54]. The 10 nm gold nanorods were functionalized with CTAB to adjust their respective lengths to 50 nm and 59 nm, thereby modifying the spectral peak positions of SPR from 900 nm to 980 nm. The nanorods were coated with PEG-Tat peptide, which was observed to be internalized by intracellular receptors and aggregated during the process of cellular uptake. The optical properties of these gold nanorods enhance spectral contrast and reduce speckle noise as an OCT contrast agent. This study illustrates the effective labeling of retinal pigment epithelial cells using gold nanorods, offering valuable information on the utilization of these agents for biomedical imaging applications in ophthalmology. In addition, the elongated structure of nanorods contributes to increased localization and amplification of the electromagnetic field. Hosseinniay et al. synthesized gold nanorods with an average diameter of 24 ± 1 nm and length of 5 ± 1 nm, yielding an aspect ratio of approximately 4.9 (Figure 4C) [76]. The longitudinal and transverse plasmon bands of the synthesized nanorods were observed at wavelengths of 740 nm and 537 nm, respectively. The nanorods were modified with a specific aptamer on their surface and utilized as biosensors for quantifying the concentrations of C-reactive protein (CRP). The interaction between proteins in the biosensor and between the aptamer and protein resulted in a shift in LSPR, attributed to changes in the refractive index of the gold nanorods. This modification enabled a selective response to BSA, TNF-α, and CRP proteins, thereby demonstrating a notable specificity towards CRP. Peixoto et al. studied the use of gold nanorods in surface-enhanced fluorescence immune biosensors, revealing the mechanisms through which their distinctive geometric properties enhance the detection efficiency of these biosensors (Figure 4D) [77]. The inclusion of bovine serum albumin (BSA) and anti-BSA in the surface modification resulted in a shift towards longer wavelengths. They emphasized that the enhancement of the longitudinal LSPR electromagnetic field could mitigate fluorescence quenching, leading to increased sensitivity and detection capabilities of the fluorescence signal. Peixoto and colleagues significantly improved the sensitivity of detection for specific biomarkers and the ability to identify low-concentration targets in biological samples.

GNPs with star-shaped structures exhibit unique characteristics as a result of their anisotropic morphology. The enhanced specific surface area, ability to significantly amplify a local electromagnetic field via sharp tips, and strong localized surface plasmon resonance (LSPR) effect improve SERS and also enhance catalytic activity and photothermal conversion efficiency [78]. These distinctive features of gold nanostars contribute to their applicability in high-sensitivity applications such as SERS imaging, medical imaging, lateral flow immunoassay, as well as in cancer diagnosis and treatment [60,63,65]. Atta group fabricated gold nanostars with sharp branches through the seed-mediated method and employed them in a lateral flow immunoassay for the detection of Yersinia pestis (Figure 5A) [60]. Gold nanostars possess a spherical core along with numerous spikes extending outwards from the core, resulting in an augmentation of the surface-to-volume ratio. This led to an increased sensitivity and rapid detection of bacterial and viral biomarkers. Gold nanostars were synthesized through the adjustment of Ag^+^ concentration, resulting in varying sizes. The samples with an average spike length of 95 ± 7 nm exhibited increased UV-vis absorption in the visible spectrum and a greater extinction coefficient in comparison to other samples, resulting in enhanced sensitivity. Zhang and colleagues synthesized gold nanostars through a one-pot method with Good’s buffers. They highlighted the benefits of these nanostars in terms of size, branch distributions, and functionalization as contrast agents (Figure 5B) [63]. These gold nanostars exhibited unique optical characteristics, with gold nanostars modified with HEPES and EPPS exhibiting LSPR with peaks at 750 nm, and those modified with 3-(N-morpholino)propanesulfonic acid (MOPS) showing LSPR peaks at 720 nm, depending on the buffer type utilized. The size of the particles was determined by Feret diameters of approximately 45 nm and exhibited various morphologies that affected the photoacoustic responses. Gold nanostars functionalized with melanin demonstrated improved imaging capabilities due to melanin’s unique optical properties. This study highlights the potential enhancement of photoacoustic signals due to the greater absorption cross-section per nanoparticle volume and SPR effect. Pearl and colleagues presented gold nanostars, identified as NSt-1 and NSt-2, for use as imaging and photothermal therapeutic tools (Figure 5C) [65]. The nanostars were fabricated using a surfactant-free method, employing 3 nm and 15 nm gold nanoparticles as seeds for NSt-1 and NSt-2, respectively. The size variability of the AuNSts contributes to their unique plasmonic absorption properties in the NIR region. Specifically, these nanoparticles displayed strong and broad absorption spectra with peaks around 600 nm for NSt-1 and 750 nm for NSt-2, depending on the number of tips present. The study also investigated the therapeutic capabilities of these nanostars in photothermal therapy by using AuNSt treatment on A549 cells followed by exposure to NIR light. This resulted in a notable temperature elevation from 20 °C to 45 °C and 49 °C for each sample type, respectively, at a concentration of 60 μg/mL. Moreover, this group analyzed the fluorescence lifetimes of the nanostars and determined that the AuNSts exhibited features corresponding to both cell autofluorescence and fluorescence originating from the nanostars. The dual fluorescence behavior of these compounds demonstrates their potential for use in advanced imaging applications.

Gold nanostars offer advantages, but their consistent and reproducible synthesis is challenging, which can affect their optical properties and functionality. To address these issues, gold nanoshells have been developed as a versatile alternative with adjustable optical properties. Gold nanoshells are characterized by a core–shell structure in which a dielectric core is surrounded by a thin layer of gold [79]. This distinctive configuration enables accurate adjustments of their SPR over a wide range of the electromagnetic spectrum, encompassing the visible and NIR regions. The resonance wavelength of gold nanoshells is influenced by the ratio between the core size and the thickness of the gold shell. This property allows gold nanoshells to efficiently absorb and scatter light efficiently at specific wavelengths, making them highly adaptable for various applications, including targeted photothermal cancer therapy and optical imaging techniques such as OCT [80,81]. In addition, gold nanoshells enable the occurrence of plasmon hybridization effects that can be adjusted by varying the thickness of the shell. These hollow structures present a unique paradigm of light interaction in contrast to solid structures, showcasing potential resonant cavity effects that are useful for specific applications. By manipulating the ratio of the core-to-shell thickness, the scattering and absorption characteristics of gold nanoshells can be tailored to improve imaging contrast. For instance, in OCT, the enhanced scattering properties of gold nanoshells allow for increased imaging resolution and enhanced penetration depths, resulting in a more precise visualization of retinal structures [82]. Wang and colleagues developed porous gold–silver (Au-Ag) alloy nanoshells that amplified the Raman signal by a factor of 68 compared to 100 nm gold nanoparticles (Figure 6A) [67]. The calculated Raman enhancement value was approximately 7804, suggesting potential applications as a carrier of cargo and a probe for SERS with high sensitivity. Chen and colleagues employed hollow gold nanoshells functionalized with biotin-PEG-SH (HAuNS@PEG-bio) as a theragnostic tool for breast cancer (Figure 6B) [83]. The strong absorbent properties of HAuNS@PEG-bio in the NRI-II, combined with its high photothermal conversion efficiency of 63%, indicate its potential for synergistic utilization in breast cancer treatment alongside photothermal and radiosensitizing therapies. Furthermore, Wang and colleagues utilized poly(ethylene glycol)-coated gold–silica nanoshells as a contrast agent for photoacoustic tomography in brain imaging (Figure 6C) [84]. The nanoshell solution was prepared with a gold shell measuring 10–12 nm in thickness, exhibiting an optical absorption peak at 800 ± 5 nm. As a consequence, there was a rise in the optical absorption of light by the blood, leading to an enhanced contrast between the blood vessels and adjacent brain tissues. Due to the increased optical absorption of nanoshells, the efficiency of visualizing brain vessels was enhanced to 63%.

Moving on from single-morphology gold nanoparticles, which primarily utilize their distinct shapes for improved SPR and light absorption and scattering effects, this paper now focuses on gold nanochains and nanoclusters. Gold nanochains are linear structures composed of connected gold nanoparticles, resulting in unique anisotropic optical and electronic properties. The interaction of plasmonic within neighboring nanoparticles in a sequence improves SPR effects, influenced by both the length of the chain and the distance between the nanoparticles [85]. This sensitivity allows for refined optical properties through manipulation of the physical parameters, making gold nanochains valuable for biosensing and molecular imaging [86]. The elongated structure and enhanced SPR can be utilized for the precise detection of biomolecules. Moreover, gold nanoclusters, with a diameter less than 2 nm, consist of a few to several hundred gold atoms and exhibit molecular-like properties, including electronic states and fluorescence. Their fluorescence, varying with size within the visible to NIR spectrum, and strong resistance to light-induced degradation, make them suitable for biomedical imaging [69]. The luminescent properties, along with their capacity to modify their surface with biomolecules, make gold nanoclusters appropriate for use as fluorescent markers in biological assays and targeted drug delivery [87]. Nguyen and colleagues synthesized chain-like gold nanoparticle (CGNP) clusters that were functionalized with arginine–glycine–aspartic acid (RGD) peptides for use as PAM and OCT contrast agents (Figure 7A) [7]. CGNP clusters are formed through the organization of femtosecond laser-created GNPs, as well as the assembly of these GNPs using the pentapeptide CALNN and cysteamine. The contrast agent demonstrated a redshift peak wavelength of 650 nm, as well as excellent biocompatibility and photostability in comparison to gold nanoparticles. These attributes allow researchers to study choroidal neovascularization in rabbit models through the use of PAM and OCT. The redshift phenomenon facilitated the distinction between the distribution of the exogenous contrast agent and the normal blood vessels, which exhibit an absorption peak around 560 nm. Furthermore, they developed ultraminiature chain-like gold nanoparticle clusters (GNCs) through the self-assembly of gold nanoparticle monomers, leveraging the unique optical and chemical properties inherent in these structures (Figure 7B) [8]. The synthesized ultraminiature GNCs show an optical absorption band that has shifted from 529 nm to 610 nm, with dimensions measuring approximately 30.0 ± 2.1 nm in length and 7.8 ± 1.1 nm in width. These GNCs exhibit improved excretion through the kidneys and decreased overall toxicity, making them well suited for in vivo applications. Their high degree of photostability and multimodal imaging capabilities, integrating photoacoustic and fluorescence imaging, offer a comprehensive assessment of choroidal neovascularization. This enables accurate identification and characterization of neovascular proliferation, early detection of diseases, and monitoring of disease progression through non-invasive methods. Therefore, it can be concluded that gold clusters also show significant potential for clinical translation.

Yang and colleagues engineered Au_44_MBA_26_ nanoclusters (Au_44_MBA_26_-NLG) for use as both a luminescent and photothermal/photodynamic component in a theragnostic probe using the NaOH-mediated NaBH_4_ reduction method (Figure 8A) [71]. This probe demonstrates luminescence in the NIR-II range at wavelengths of around 1080 nm and 1280 nm, with a size of approximately 2.7 nm, enabling better tissue penetration and imaging quality with minimized background interference. Furthermore, the unique structure of these nanoclusters enables them to perform two distinct roles, enabling precise imaging in the NIR-II spectrum and triggering photothermal and photodynamic therapeutic reactions upon stimulation. This multifunctional approach highlights the potential of nanoclusters in advancing cancer treatment through the integration of accurate diagnostic imaging and efficient localized therapy. Hada et al. investigated gold nanoclusters capped with glutathione (GSH-AuNCs) as a contrast agent for confocal fluorescence imaging (Figure 8B) [72]. The contrast agent demonstrated distinctive optical properties, including strong NIR fluorescence emission, effectively addressing the issue of limited penetration depth seen in existing fluorescence probes and high autofluorescence. They demonstrated a 9.9% rise in the quantum yield of NIR emission, indicating their potential for effective use in deep tissue imaging applications.

Our study of the synthesis and categorization of gold nanoparticles demonstrates that variations in size and shape, ranging from nanospheres to nanochains, significantly influence their unique optical and electronic properties. These characteristics subsequently influence their utilization in diverse fields. Moreover, each morphology type, such as symmetrical nanospheres, elongated nanorods, elaborately branched nanostars, or hollow nanoshells, inherently possesses distinct advantages and disadvantages. The unique characteristics of nanoparticles, including their shape, size, and surface modifications, can influence their optical and physicochemical properties, contributing significantly to the enhancement of absorption, scattering, SPR, and heat conversion efficiency. Therefore, it is crucial to comprehend the influence of the physical attributes of GNPs to optimize their utility as key components in biomedical engineering applications, including imaging, diagnostics, and therapeutics.

### 2.3. Criteria for Selection of Gold-Based Contrast Agents in Retinal Imaging

Selecting the appropriate GNPs for retinal imaging necessitates careful consideration of various criteria to optimize imaging outcomes. Gold-based contrast agents are significant in improving retinal imaging by enhancing the contrast levels between eye structures, thus facilitating the identification of ocular abnormalities and diseases. To serve as effective contrast agents, the features of GNPs may vary depending on the imaging technique and specific visualization requirements of ocular structures, while still possessing essential properties [88].

The optical properties of GNPs are influenced by their size through SPR, impacting their absorption specifically in the NIR region. NIR light can penetrate further into retinal tissues, thereby enabling improved visualization of deeper ocular structures. The ideal size range for GNPs to exhibit efficient SPR properties for in vivo imaging is typically between 10 nm and 100 nm. In addition to their optical properties, the size of GNPs plays a significant role in determining their biological behavior, such as toxicity and circulation within the bloodstream. Nanoparticles with dimensions smaller than 10 nm are capable of being efficiently eliminated from the body via renal excretion, thereby reducing the risk of systemic toxicity [89]. However, their diminutive dimensions may result in the penetration of cellular membranes, potentially leading to cytotoxic effects if not adequately stabilized [90]. On the other hand, nanoparticles that exceed 100 nm in size may exhibit prolonged retention within the body, thereby enhancing longitudinal imaging capabilities; however, this phenomenon also prompts potential concerns regarding long-term toxicity and accumulation over time. These particles are primarily sequestered by the liver and spleen as a result of phagocytosis by the reticuloendothelial system [91]. Furthermore, it is essential for GNPs to remain in circulation in the bloodstream for an extended period to effectively reach targeted retinal sites for the purpose of capturing comprehensive images of the retinal blood vessels and related pathologies. To prevent GNPs from eliciting notable immune responses or inducing harm to cells, it is essential to rigorously regulate their shape, size, and surface chemistry through the use of non-toxic materials and surface modifications that are non-immunogenic. For instance, applying coatings of biocompatible and biologically active layers like PEG can improve their biocompatibility. PEGylation decreases protein adsorption and opsonization, thereby reducing the recognition and uptake of these nanoparticles by the immune system, prolonging their circulation time in the bloodstream, and minimizing potential side effects [92]. 

In addition, the utilization of GNPs with specific ligands such as antibodies, peptides, or small molecules that target unique cellular markers or receptors associated with retinal diseases can be considered for enhancing the specificity and efficacy of imaging. This is critical for identifying and visualizing damaged or anomalous cells in conditions such as AMD, choroidal or retinal neovascularization, and diabetic retinopathy. For instance, GNPs can be modified with anti-vascular endothelial growth factor (anti-VEGF) antibodies to selectively target vascular endothelial growth factors. VEGF plays a crucial role in the pathological angiogenesis associated with these diseases, and targeting it can offer valuable insights into disease progression and treatment responses [93]. Moreover, surface modifications of GNPs can involve the incorporation of dyes or other imaging agents. These modifications are intended to enhance the detectability of GNPs in different imaging techniques, including OCT or PAM. By enhancing the optical properties of GNPs, these surface modifications enable the achievement of clearer and more intricate visual representations of the retinal structure. These enhancements aid in precise diagnosis and monitoring of retinal diseases, enhancing the effectiveness of the imaging process and providing reliable tools for assessing disease progression and therapeutic intervention efficacy. 

Furthermore, the enhanced permeability and retention (EPR) effect can be carefully considered. This phenomenon allows GNPs to accumulate preferentially in tissues with abnormal, leaky vasculature of retinal diseases, significantly improving imaging contrast and clarity. By utilizing the EPR effect, GNPs can improve the visualization of retinal pathologies, facilitating more precise disease diagnoses and progression assessments. This targeted accumulation of GNPs in affected areas not only reduces potential side effects throughout the body and improves the safety profile of the diagnostic tool but also boosts the possibilities for theragnostic applications. The use of GNPs enables clearer imaging of specific retinal conditions and targeted delivery to the site of pathology with reduced adverse effects.

Ultimately, the important factors of size, surface modification, and biocompatibility must be considered alongside ongoing monitoring and safety assessments to guarantee the efficacy and safety of GNPs for retinal imaging. It is essential to monitor the interactions and stability of GNPs within the body system as they are utilized. This monitoring includes assessments of the interactions between GNPs and biological tissues, with particular emphasis on precision and safety. By closely monitoring these interactions, any potential negative consequences can be detected and dealt with in a timely manner to mitigate further complications. Furthermore, assessing the stability of GNPs aids in elucidating their degradation or accumulation within the biological system over time, thus ensuring that their administration does not pose any risks of long-term health complications. These safety evaluations are essential for validating the ongoing appropriateness of GNPs for clinical use, guaranteeing their ability to provide diagnostic and therapeutic benefits without posing a risk to health. These criteria emphasize the significance of the physicochemical properties of GNPs and underscore the necessity for a comprehensive framework to evaluate their long-term effects on the body system in clinical applications as well.

## 3. GNPs in Optical Imaging Techniques for Retinal Applications

This section introduces various studies pertaining to the essential characteristics of GNPs necessary for their utilization in retinal imaging applications.

Raveendran and colleagues synthesized gold nanocages (AuNcgs) that demonstrate a shifted LSPR peak towards the NIR region due to the conversion of solid silver nanocages (AgNcbs) into hollow AuNcgs (Figure 9A) [94]. The AgNcbs demonstrate a consistent wall thickness of 5 ± 2 nm and an average edge length of approximately 65 nm. Researchers utilized prepared AuNcgs in enucleated porcine eye models, followed by injections of AuNcgs to replicate conditions of uveal melanoma. They chose an interrogation region measuring approximately 0.35 by 0.35 mm^2^ for imaging purposes, with a pulse energy of approximately 71 μJ, which is below the maximum permissible exposure (MPE) levels, as recommended by the American National Standard Institute (ANSI) to ensure safety from ocular exposure. The transient and localized thermal effects induced by the laser pulses minimized the risk of thermal damage. The use of AuNcgs further enhances safety by improving the optical absorption efficiency, leading to a reduction in the necessary laser energy for optimal imaging. The thorough analysis of contrast agents, imaging modality, and setup ensured the safe application of photoacoustic imaging for retinal imaging while maintaining tissue integrity. The acquired photoacoustic and ultrasound images indicated that the AuNcgs yielded an improved contrast and resolution, allowing for more precise delineation of the melanoma regions. The enhanced PA signal obtained with AuNcgs was approximately 50.8% higher compared to signals measured without the presence of these nanostructures. This study highlighted the capacity of AuNcgs to improve the diagnostic capabilities of photoacoustic imaging for retinal diseases. In addition, Nguyen et al. demonstrated a substantial enhancement in the performance of PAM when using gold nanorods conjugated with RGD ligands (GNR-RGD) (Figure 9B) [95]. The mean dimensions of GNRs were calculated as 34.17 ± 4.85 nm in width and 103.28 ± 12.07 nm in length, which is within the appropriate range for their efficacy as a contrast agent. The unique optical properties of gold nanorods facilitated a noteworthy enhancement in the contrast and resolution of PAM images. The PAM images of the CNV were obtained using the excitation wavelengths of 578 nm and 700 nm to visualize the retinal microvasculature and the localization of GNR-RGD within the CNV. The PAM images were acquired through raster scanning at a resolution of 256 by 256 pixels in 65 s. The lateral resolution was estimated to be 4.1 μm, while the axial resolution was estimated to be 37.0 μm. The laser energy was measured at approximately 80 nJ, demonstrating a level approximately 50% below the ANSI safety limit. Consequently, the utilization of GNR-RND leads to a 27.2-fold enhancement (at 48 h post-injection) in the signal-to-noise ratio, enabling the in-depth visualization of choroidal neovascular networks. This enhancement is beneficial for the early detection and monitoring of CNV, as it enables accurate identification of abnormal blood vessel formation in the choroid. The enhanced signal amplification provided by GNR-RGD in PAM highlights its potential to improve the diagnostic capabilities of this imaging modality. In addition, they also reported noteworthy improvements in OCT imaging. The wavelengths of 846 nm and 932 nm were utilized to induce a coherence signal for stimulating the samples. The resolution was measured to be 3.8 μm for lateral and 4.0 μm for axial resolution. The B-scan OCT images were obtained at a resolution of 512 by 1024 A-lines in 103 ms at an acquisition speed of 36 kHz. Under this setup, GNR-RGD markedly increased the OCT signal, leading to a 171.4% enhancement in signal intensity. This notable enhancement in the signal-to-noise ratio enabled the thorough visualization of the vascular structures of the eye, resulting in clearer and more precise images of the CNV. This outcome emphasizes the potential of GNR-RGD in augmenting the diagnostic capabilities of OCT. Enhancements in contrast and resolution in OCT imaging are essential for the early detection and monitoring of ocular diseases such as CNV, thus highlighting the potential of GNR-RGD as a promising contrast agent for clinical applications.

Moreover, Nguyen et al. developed highly pure chain-like gold nanoparticle clusters (CGNP clusters) with absorption properties shifted towards the red end of the spectrum for the purpose of visualizing retinal microvasculature and CNV (Figure 10A) [7]. The CGNP clusters were conjugated with RGD to promote enhanced cell adhesion to the extracellular matrix. The CGNP clusters functionalized with RGD exhibit an average length of 64 nm and a width of 20 nm, accompanied by a redshift in the absorption peak at 650 nm. The resolution of the PAM system was quantified as 4.1 μm for lateral resolution and 37.0 μm for axial resolution, demonstrating adequate capability for the detection of microvasculature, capillaries, and CNV. The images were acquired with a laser pulse fluence averaging approximately 0.01 mJ/cm^2^ at wavelengths of 578 and 650 nm, equivalent to half of the maximum permissible single laser pulse fluence on the retina set by ANSI standards. Images were captured within 65 s over a region measuring 4 by 4 mm^2^ with a resolution of 256 by 256 pixels. After the rabbits were injected intravenously with CGNP cluster–RGD, the PA signal increased by up to 17-fold. This allowed the PA images to show significant contrast and strong optical absorption of the CGNP cluster–RGD against the adjacent blood vessels. This study showed that CGNP clusters combined with RGD peptides can be effective as contrast agents for PAM imaging, with the potential to enable more precise quantitative measurement and reduce inaccuracies in assessing contrast agent concentrations in extravasation. The study showed that CGNP cluster–RGD proved to be effective as an OCT contrast agent when excited by laser wavelengths of 805 nm and 905 nm. The resolution of OCT was quantified as 3.8 μm laterally and 4.0 μm axially. CGNP cluster–RGD significantly increased the contrast-to-noise ratio (CNR) of retinal tissue from 1 to 1.76, thereby enhancing its backscattering properties. Superluminescent diodes emitting at wavelengths of 846 nm and 932 nm were utilized to capture B-scan OCT images. The acquisition process involved 512 by 1024 A-lines and an acquisition rate of 36 kHz, resulting in images obtained within 103 ms. The findings of this study demonstrate the efficacy of CGNP cluster–RGD for visualizing CNV in live rabbits.

Nguyen et al. conducted a study on the detection of CNV progression across different age groups after receiving bevacizumab treatment using a PAM and OCT integrated system (Figure 10B) [96]. PAM involves generating a photoacoustic signal from an optical parametric oscillator and focusing it on the corneal surface. Molecular imaging was enhanced through the utilization of gold nanoparticles bound to RGD peptide, leading to improved sensitivity in PA imaging. This enhancement was evidenced by the specific targeting of CNV lesions, as the nanoparticles accumulated at the affected sites. PAM images were obtained employing excitation wavelengths of 578 nm for the visualization of choroidal and retinal vascular networks, and 650 nm for the detection of nanoparticles localized at CNV sites. The enhanced PA signal from nanoparticles allowed for notable contrast and detailed visualization of CNV, with a 17-fold increase in PA signal observed post-injection. Three-dimensional PAM images were reconstructed without post-image processing, providing detailed information on the volume and depth of blood vessels. After 28 days of treatment with bevacizumab, PAM images revealed changes in the retinal structure and choroidal blood vessels, with minimal PAM signal observed in the group of 4-month-old rabbits. Histological analysis verified the presence of newly developed CNV below the neurosensory retina, consistent with the PAM result. Furthermore, the OCT system is adapted to achieve imaging depths of up to 1.9 mm. To enhance the sensitivity and quality of OCT, gold nanoparticles conjugated with RGD are employed. Cross-sectional B-scan and 3D volumetric OCT images were acquired at various intervals, showing the initial OCT signal of retinal blood vessels and CNV. Following the injection of nanoparticles, there was a significant increase (176%) in the OCT signal from the CNV, reaching its peak at 48 h post-injection and subsequently decreasing gradually. Prior to receiving bevacizumab treatment, the group of 4-month-old rabbits experienced a notable reduction in CNV of approximately 82.83% and a decrease in the normalized CNV area. However, there was no notable discrepancy noted in the 14-month-old group, suggesting the limited influence of bevacizumab on aged rabbits. The study demonstrates that the integrated PAM and OCT system, along with gold nanoparticles conjugated with RGD, can improve the detection and monitoring of CNV progression and treatment effectiveness. This method provides detailed, non-invasive imaging capabilities, which enhances its utility in studying CNV and assessing therapeutic treatments.

In addition, they have tailored the size of 7–8 nm ultraminiature gold nanoparticle clusters (GNCs) to allow for renal clearance and excretion through urine, thus addressing biosafety concerns (Figure 11) [8]. The unique chain-like structure of the material resulted in a redshifted optical absorption peak at 610 nm, thereby enhancing its optical absorption and scattering properties. Furthermore, they conjugated the chain-like ultraminiature GNCs with RGD peptides to target angiogenesis. The study showed that ultraminiature GNCs exhibited markedly increased PA and OCT signals when compared to larger GNCs. At a mass concentration of 250 μg/mL, the ultraminiature GNCs increased the PA signal by 3.03-fold and 1.99-fold over the large nanoparticles at 100 μg/mL and 250 μg/mL, respectively. The nanoparticles demonstrate significant resistance to light-induced degradation and are capable of producing clear images of retinal blood vessels and CNV with high differentiation, maintaining their efficacy for up to three weeks after injection when stimulated with light at wavelengths of 578 and 700 nm. This study emphasizes that the surface-to-volume ratio of the nanoparticles has a significant impact on the PA signal, as smaller nanoparticles produce a stronger signal because of greater surface area exposure. In addition, at 100 μg/mL, the OCT intensity of ultraminiature nanoparticle clusters was 1.65 times lower than that of large GNCs. However, this difference diminished with increased concentrations. Although they are small in size, the ultraminiature GNCs still offer significant signal enhancements, thus making them ideal for high-resolution imaging of retinal vasculature and CNV. The study exhibited that the ultraminiature GNCs could effectively facilitate detailed imaging of CNV, providing adequate contrast for identifying anatomical structures. In conclusion, the study showcases the capability of ultraminiature GNCs to serve as adaptable and efficient contrast agents for multimodal molecular imaging. The renal clearance of the clusters elucidates potential concerns regarding long-term toxicity that may arise from conventional larger gold nanoparticles. Furthermore, this study represents a noteworthy progression in biomedical imaging by providing safer and more effective methods for diagnosing and monitoring ocular and other diseases.

Song et al. designed gold nanodisks (GNDs) with enhanced scattering capabilities compared to gold nanorods, with a resonant wavelength of 830 nm in the NIR region that aligns with the light source utilized in OCT (Figure 12A) [97]. These GNDs have a diameter of 160 nm and exhibit strong optical signals in OCT, even at low concentrations of 1 pM, making them highly sensitive contrast agents for retinal imaging. They demonstrated their capacity to bind to vascular endothelial growth factor (VEGF), thereby inhibiting VEGF-induced migration of endothelial cells. The high scattering properties of GNDs enhance the backscattering of light signals in OCT, resulting in enhanced visibility and resolution of retinal structures. This study also showed that intravitreally injected GNDs can suppress retinal neovascularization, indicating their potential as both imaging and therapeutic agents for the treatment of diseases like oxygen-induced retinopathy. The small size and optimal surface properties of GNDs contribute to low toxicity and rapid clearance from the vitreous, supporting their safety and efficacy in clinical applications.

Chemla and colleagues used GNPs as contrast agents for cell tracking based on their absorbance and scattering properties (Figure 12B) [98]. This study enhanced OCT imaging contrast using GNPs. The signal intensity of cells labeled with GNPs exhibited a statistically significant increase when compared to cells without labeling. Specifically, the average signal intensity increased from 2.2-fold to 4.3-fold with an increase in the concentration of GNPs. OCT images revealed clusters of photoreceptor precursor cells situated in the subretinal space, with the amplification of OCT signals by GNPs facilitating the visualization of small cellular clusters migrating from the injection site in the subretinal site towards the inner retinal layers. Furthermore, they functionalized GNPs with Rhodamine, a fluorescence dye, to improve the fluorescence visualization of transplanted retinal cells. The transplanted photoreceptor precursors, marked with both green fluorescent protein and GNPs, were tracked through fluorescence fundus imaging. Consequently, the GNPs remained contained within the cells without displaying any toxicity or fluorescence quenching over the observation period. Fluorescence images provided high-resolution tracking of smaller cellular aggregations. This facilitated the accurate identification of cell clusters within the retina for a duration of up to 30 days following transplantation, highlighting the capability of GNPs for long-term, detailed monitoring of cell migration and viability in retinal treatments.

Furthermore, Nguyen et al. developed GNPs labeled with indocyanine green (ICG) and conjugated with RGD peptides (ICG@CGNP cluster–RGD) for the purpose of improving multimodal imaging techniques such as PAM, OCT, and fluorescence microscopy [99]. These nanoparticles exhibit distinctive optical properties with a redshift absorption peak at 650 nm, while maintaining the small size of GNPs. The synthesis process entails the self-assembly of 20 nm spherical GNPs into chain-like clusters, PEGylation for improved stability and biocompatibility, and conjugation with RGD peptides to target integrin receptors overexpressed on activated endothelial cells and neovascularization. These ICG-labeled and RGD-conjugated CGNP clusters exhibit high photostability, low cytotoxicity, and enhanced imaging contrast in both in vitro and in vivo models. In rabbit models of CNV and RVO, these clusters have successfully targeted diseased tissues, providing high-contrast images that facilitate early diagnosis and monitoring of retinal diseases. This multifunctional approach signifies a substantial progression in molecular imaging, offering a promising tool for the comprehensive visualization of retinal pathologies.

In conclusion, GNPs have become a significant factor in advancing optical imaging techniques, particularly for retinal applications. The distinctive optical properties of GNPs, including their capacity to absorb and scatter light and their ability to be modified with different ligands and dyes, enable them to function effectively as contrast agents. These developments have contributed to notable improvements in various imaging techniques such as PAM, OCT, and fluorescence microscopy, leading to enhanced resolution and clarity in images. Moreover, GNPs have demonstrated significant potential as therapeutic tools, making new opportunities for treating various retinal illnesses. With the progression of molecular imaging technology, the utilization of nanoparticles is anticipated to have a significant impact on the timely identification, management, and treatment of visual disorders. This approach will enhance our understanding of the intricate pathological mechanisms associated with ophthalmic diseases and facilitate the development of more efficient treatment approaches.

Although there is significant promise in utilizing GNPs for retinal imaging, there are still various obstacles that need to be overcome. A thorough investigation is necessary to address critical factors such as long-term safety, biocompatibility, and optimal dosage. The development and clinical approval of GNP-based agents requires thorough evaluation and validation to ensure their safety and efficacy in human applications. Nevertheless, the notable outcomes attained thus far underscore the substantial potential of GNP-based imaging methods for the timely identification and surveillance of retinal disorders. In the future, ongoing research, innovation, and development will be crucial in realizing the full potential of GNPs in clinical applications.

## 4. Clinical Applications and Challenges of GNPs in Retinal Imaging

GNPs have shown promising potential in improving retinal imaging using different modalities, leading to enhanced resolution and contrast. Many studies have investigated the therapeutic use of imaging applications, emphasizing their multifunctional capabilities. However, several challenges must be addressed to facilitate the safe and effective utilization of these nanoparticles in medical applications following their transition from research settings. This manuscript explores the important factors of clinical utilization and the challenges related to retinal imaging with GNPs. The key subjects of discussion encompass safety and toxicity issues, methods to address existing imaging limitations, enhancements for clinical applications, and the potential for advancement and regulatory aspects of GNP-based imaging technology.

This study will offer a comprehensive analysis of the integration of GNPs into standard clinical settings. This section addresses specific challenges and considerations specific to the clinical translation of GNPs for retinal molecular optical imaging, essential for a thorough understanding of the topic. While the regulatory processes for medical substances are widely recognized and follow the standard rules, GNPs pose challenges due to their unique properties. This detailed discussion offers valuable perspectives on the essential processes and factors involved in the incorporation of GNPs into clinical settings. It is essential to focus on these issues to improve the utilization of GNPs in retinal imaging and to address the associated challenges effectively.

### 4.1. Safety and Toxicity

Ensuring the safety and biodistribution throughout the body and elimination pathways of GNPs is paramount for their clinical applications, necessitating rigorous assessment of their toxicity profiles. After being administered systemically, GNPs are distributed throughout the body, with the spleen and liver identified as the main sites of accumulation, followed by the lungs and kidneys [100]. The biodistribution of nanoparticles within biological systems is notably impacted by their dimensions, morphology, surface properties, and modifications. Nanoparticles below 6 nm in size can penetrate the renal filtration barrier and undergo renal excretion, leading to shorter circulation periods and rapid clearance from the bloodstream. On the other hand, larger GNPs are primarily absorbed by the liver and spleen through the mononuclear phagocyte system, facilitated by macrophages in these organs that identify and engulf the nanoparticles. The morphology of GNPs also influences their biodistribution as a result of their unique surface characteristics and interactions with cell membranes. The presence of a surface charge on GNPs affects their interaction with serum proteins and cell membranes, wherein positively charged nanoparticles have a greater tendency to bind to cell membranes, potentially resulting in a higher cellular uptake, while negatively charged or neutrally charged nanoparticles may persist longer in the bloodstream [101]. In addition, surface modification can appreciably impact the distribution of GNPs in biological systems. The addition of targeting ligands, such as peptides or antibodies, to functionalize GNPs can guide them to specific cells or tissues, potentially reducing off-target accumulation and improving therapeutic effectiveness [102]. For instance, PEGylation has the capability to improve the longevity of nanoparticles in the bloodstream and their overall stability through the reduction in opsonization and subsequent uptake by the mononuclear phagocyte system.

The mechanisms for clearance are essential in assessing the safety and effectiveness of GNPs in clinical applications. Nanoparticles below the size limit for renal filtration are mainly excreted through the kidneys, passing through the glomerulus, and then removed in urine, minimizing long-term exposure and potential harm. The liver primarily clears larger nanoparticles through the processing and excretion by hepatocytes and Kupffer cells into the bile for elimination via feces [103]. The spleen functions to capture and filter foreign materials from the bloodstream, as splenic macrophages retain GNPs for prolonged periods due to their slow metabolism and removal from the body.

It is crucial to comprehend the biodistribution and clearance patterns of GNPs to evaluate their safety and toxicity. The accumulation of nanoparticles in the liver and spleen poses potential risks of hepatic and splenic toxicity, especially when larger doses are administered. Extensive investigations are required to ensure biocompatibility and minimize adverse effects due to the chronic exposure and long-term retention of GNPs in these organs. Moreover, it is essential to thoroughly assess the effects of surface coatings and functionalization on biodistribution and clearance when aiming to improve therapeutic efficacy or targeting. For example, while PEGylation can decrease immune response and extend the duration of circulation, it can also impact the processes and rates of excretion. Comprehensive studies on the biodistribution, clearance, and long-term effects of GNPs are imperative to guarantee their safe transition from research to clinical use. Additional research should prioritize the enhancement of the design of GNPs to achieve a balance between effectiveness and safety, taking into account variables such as size, shape, surface charge, and functionalization, to attain the desired diagnosis and therapeutic outcomes while minimizing toxicity.

### 4.2. Overcoming Limitations of Imaging Systems

While GNPs have demonstrated their ability to improve image resolution and contrast, it is imperative to address the existing limitations of the imaging system to facilitate their utilization in clinical settings. This section discusses the primary obstacles and potential solutions to address these constraints.

PAM exhibits high spatial resolution and excellent contrast but faces challenges with limited penetration depth, particularly in deeper tissues. Potential technological improvements to the PAM system may include the development of imaging systems with increased penetration capabilities into tissues, while still preserving a high degree of resolution and contrast. An alternative method could entail utilizing various nanoparticle variants capable of enhancing contrast at increased depths. The integration of multimodal imaging modalities could offer a more thorough understanding of tissue characteristics. OCT provides detailed imaging of tissue microstructures; however, it encounters difficulties with penetrating to significant depths due to the scattering and absorption of light in biological tissues. Improvements such as adaptive optics, enhanced light sources, and the integration of OCT with other modalities, such as PAM, have the potential to address these constraints and yield higher-resolution images at increased tissue depths. Fluorescence microscopy is hindered by photobleaching and phototoxicity, resulting in the degradation of image quality and potential damage to biological tissues. Enhancing the properties of fluorophores and refining imaging procedures can minimize these challenges, enabling extended and more comprehensive imaging.

Obtaining an SNR is essential for precise imaging. The presence of non-specific binding and autofluorescence in biological tissues may hinder the detection of signals from GNPs. Increasing the specificity of GNP targeting and utilizing sophisticated image processing algorithms can greatly enhance the SNR, resulting in clearer and more reliable images. However, the necessary instruments for imaging systems are frequently costly and intricate, which hinders their extensive implementation in clinical adoption. The consideration of developing a cost-effective and user-friendly imaging system is also necessary. This includes streamlining the technology to increase accessibility and affordability, as well as establishing extensive training programs for clinicians to facilitate its integration into clinical practice.

### 4.3. Enhancing GNPs for Clinical Use in Retinal Imaging

Enhancing the properties of GNPs for clinical application in retinal imaging is necessary to improve their effectiveness, safety, and specificity. This section focuses on strategies to enhance GNPs and discusses their design, functionalization, and utilization in multimodal imaging techniques to achieve clinical outcomes. By leveraging advanced design and functionalization methods, GNPs can be tailored to meet the specific requirements of retinal imaging, ensuring improved diagnostic and therapeutic capabilities.

It is crucial to optimize the size and shape of GNPs to ensure their effectiveness in retinal imaging. Smaller nanoparticles can be quickly eliminated through the kidneys, leading to a decrease in potential harm. Although providing improved optical characteristics, larger nanoparticles have a tendency to gather in the liver and spleen. The morphology of GNPs is also a key factor in cellular internalization and contrast enhancement for imaging purposes. The stability and biocompatibility of GNPs were enhanced through surface coating, resulting in reduced opsonization and immune recognition. Functionalizing GNPs with specific targeting ligands has the potential to increase their affinity for retinal tissues, leading to improved accuracy in imaging and therapeutic applications.

The integration of GNPs with PAM can noticeably enhance imaging contrast and resolution. By optimizing the size and coating of GNPs, the penetration depth and signal intensity of PAM can be improved. Advanced laser systems and image processing algorithms further augment the capabilities of PAM. However, PAM has not yet obtained approval from the U.S. Food and Drug Administration (FDA) for clinical use, underscoring the necessity for additional evaluation and validation. GNPs can be utilized to improve the contrast and specificity of OCT. Tailoring the optical characteristics of GNPs can achieve better distinction between retinal layers and pathological attributes. The integration of OCT with PAM offers additional information that enhances understanding of both the structure and function of retinal tissues. Fluorescence microscopy is enhanced by the robust fluorescent properties of GNPs. Overcoming issues like photobleaching and phototoxicity requires the refinement of durable fluorophores and enhancement of imaging protocols. Functionalizing GNPs with fluorophores known for their high photostability and high-intensity emission may improve the efficacy and longevity of fluorescence imaging.

In clinical applications, GNPs have the potential to serve dual purposes in imaging and therapy, a concept known as theranostics. These multifunctional GNPs can be engineered to deliver therapeutic substances and also function as contrast agents for imaging, which can enhance the accuracy of treatment effectiveness and monitoring of disease progression. By engineering GNPs to incorporate drug molecules within their structure or to attach them onto the surface, targeted drug delivery can be implemented, providing advantages for conditions like AMD and diabetic retinopathy. The dual functionality enhances diagnostic precision and therapeutic outcomes by enabling localized treatment with reduced systemic exposure and side effects.

The distinctive optical characteristics of GNPs can be utilized for multimodal imaging, leveraging the advantages of various imaging techniques to gain comprehensive insight into disease status and progression [1,7,104]. In addition, GNPs possess the distinct capability to absorb and convert NIR light into thermal energy, making them appropriate for use in photothermal therapy (PTT) [105,106]. PTT involves irradiating GNPs with NIR light, leading to localized heating that can selectively destroy cancer cells or other pathological tissues. GNPs possess a high photothermal conversion efficiency and the ability to be precisely targeted towards diseased sites, thereby minimizing damage to adjacent healthy tissues [107,108]. This dual function enables the real-time monitoring of therapeutic response, facilitating the development of personalized and adaptive treatment approaches.

Furthermore, GNPs can play a role in delivering a range of therapeutic substances, such as medications, proteins, and nucleic acids [2,109]. The sizable surface area of these materials enables a high loading capacity, and their surface can be altered for precise and targeted delivery. This enhances the effectiveness of therapy while reducing unintended side effects. The therapeutic efficacy and biodistribution of GNP drug delivery systems can be observed through imaging, providing an additional dimension of theragnostic functionality. The versatile features of GNPs enable integrated diagnosis and therapy, advancing personalized medicine in clinical applications.

### 4.4. Clinical Translation and Regulatory Considerations

The translation of GNPs from research to clinical settings involves navigating several phases such as preclinical testing, clinical trial, and obtaining regulatory authorization [110], which are critical for ensuring their safe and effective use. Every stage of the process poses its own challenges that need to be addressed to guarantee the safety and effectiveness of GNPs in clinical applications.

Preclinical testing, which is also referred to as preclinical development, thoroughly assesses the safety, efficacy, and biocompatibility of GNPs through in vitro and in vivo studies. These studies are crucial for comprehending the biodistribution, clearance mechanisms, and possible toxicity of GNPs. Thorough preclinical data are essential for forecasting human responses and identifying any adverse effects at the early stages of development. This stage emphasizes the assessment of the cytotoxicity, genotoxicity, and immunogenic properties of GNPs, as well as their interactions with cellular systems and the potential induction of adverse immune reactions. Long-term studies aid in comprehending the enduring impacts of GNP accumulation in vital organs such as the liver and spleen. In addition, the preclinical efficacy studies demonstrate the therapeutic potential of GNPs in drug delivery and photothermal therapy. The studies reveal how GNPs enhance imaging contrast and efficiently deliver therapeutic agents to target tissues.

Following successful preclinical testing, GNPs must proceed to clinical trials involving several stages (Phase I to III) to confirm their safety and efficacy in humans. Clinical trials gather data on the pharmacokinetics, pharmacodynamics, and general clinical effectiveness of GNPs. Phase I trials consist of a limited number of volunteers who are either healthy or patients, aiming to assess the safety, tolerance, and appropriate dosage of GNPs, while closely monitoring for any immediate adverse reactions. Phase II trials are conducted on a larger patient population in order to evaluate the effectiveness of GNPs for specific therapeutic or diagnostic purposes, with a focus on safety monitoring and dosing regimen optimization. Phase III trials, conducted on a significantly larger scale and longer duration than Phase I or II, provide extensive data on the efficacy and safety of GNPs relative to standard treatments or placebos, which is essential for obtaining regulatory approval.

Obtaining regulatory approval from agencies such as U.S. FDA or the European Medicines Agency (EMA) entails an intricate and rigorous process. This stage verifies that GNPs and PAM adhere to all safety, efficacy, and quality standards prior to their implementation in clinical usages. Comprehensive documentation of preclinical and clinical trial data is necessary to establish regulatory compliance for GNPs and PAM. This includes comprehensive reports on manufacturing processes, quality assurance practices, and hazard evaluations. Continual monitoring for both GNPs and PAM is essential in post-market surveillance to track their long-term safety and effectiveness in the general population. This entails the reporting of any negative occurrences and implementing essential modifications to usage guidelines or formulations.

The process for obtaining regulatory approval for imaging agents based on GNPs is intricate and stringent, requiring thorough preclinical and clinical evaluation to establish safety, effectiveness, and quality. Understanding and complying with these regulatory requirements necessitates a significant investment of time, resources, and specialized knowledge. Moreover, it is crucial to standardize the synthesis and characterization of GNPs in order to guarantee consistent and reproducible results in various studies and applications. The validation of the imaging performance and safety of GNPs in extensive clinical trials is essential for obtaining regulatory endorsement and clinical adoption.

The implementation of this approach in clinical practice presents logistical and training obstacles. Healthcare professionals should undergo training to efficiently utilize new imaging systems and correctly interpret the findings. It is essential to showcase the clinical advantages and cost efficiency of imaging techniques based on GNPs to facilitate their acceptance and utilization. The interoperability of GNPs in different healthcare systems must be considered, ensuring the seamless integration of GNP-based technologies into existing healthcare infrastructure. This encompasses the alignment with existing imaging systems, user-friendly interface for healthcare professionals, and the capacity to interpret results generated by GNP-based imaging technologies accurately.

In addition, the incorporation of GNPs into clinical practice necessitates careful examination of ethical, legal, and societal implications. This encompasses elements such as obtaining informed consent, safeguarding patient privacy, ensuring cost-effectiveness, and promoting equitable access to this technology. Informed consent is imperative, as patients must have a comprehensive understanding of the potential benefits and risks associated with the utilization of GNPs. Given the potential economic burden on hosts of GNP-based therapies, it is important to establish strategies that promote equitable access to this technology among various socio-economic populations. Because GNPs represent a new development in medical diagnostics and treatments, it is essential for healthcare providers to receive proper training to use these nanoparticles effectively and safely.

In conclusion, the process of achieving clinical translation and regulatory approval for the use of GNPs is complex and involves multiple facets. This entails not only establishing the safety and effectiveness of GNPs through thorough scientific testing but also addressing the legal, ethical, societal, and practical issues associated with their utilization. By addressing these challenges directly, we can facilitate the successful integration of GNPs into routine clinical practice, thereby improving the quality and efficacy of retinal imaging. Addressing the current limitations necessitates a thorough approach that includes ongoing research, innovation, strategic partnerships, and rigorous testing. It is imperative to conduct comprehensive preclinical and clinical evaluations, adhere to regulatory requirements, and consider ethical and societal implications in the steps. Achieving successful clinical translation of GNPs can lead to significant advancements in personalized medicine and improved diagnostic and treatment outcomes.

## 5. Conclusions

GNPs have proven to be a versatile and powerful tool in retinal imaging, providing notable improvements in resolution, contrast, and multifunctional capabilities. This review has explored the different applications of GNPs for retinal applications with different morphology, which encompass their synthesis, properties, and challenges encountered in the transition of GNPs into clinical retinal imaging.

Section 1 provided an overview of the growing significance of GNPs in medical imaging, particularly in the context of retinal applications. Section 2 introduced an overview of GNPs and their physical, chemical, and optical properties, highlighting the unique features of GNPs based on different morphologies that make them suitable for use as a contrast agent. In addition, the importance of selecting appropriate gold-based contrast agents in retinal imaging was discussed, focusing on parameters such as size, shape, surface charge, and functionalization to improve imaging quality. Section 3 explored the use of GNPs in various optical imaging techniques for applications related to the retina. This section illustrated the advancements in imaging modalities, including PAM, OCT, and fluorescence microscopy, achieved using GNP-based contrast agents. This enhancement results in higher imaging quality and more detailed information. Section 4 focused on the practical applications and limitations encountered in the utilization of GNPs for retinal imaging. The safety and toxicity section examined the distribution and elimination of GNPs, emphasizing the necessity of conducting thorough safety assessments to mitigate potential risks. The following section discussed advancements in imaging technology, aimed at overcoming current limitations in imaging techniques and improving the effectiveness and safety of GNPs for clinical applications. The subsequent section outlined the detailed process from preclinical testing to obtaining regulatory approval, emphasizing the importance of rigorous testing, standardization, and the ethical and societal implications to be considered.

In conclusion, the utilization of GNPs in retinal imaging represents a noteworthy advancement in medical diagnostics and treatment. The unique properties of GNPs make them effective for use in theragnostic applications. However, the choice of nanoparticle morphology and synthesis method should be carefully considered based on the specific characteristics or needs of the tissues and molecules to be detected. For instance, considering the optical properties of each gold nanoparticle, nanorods can be used for molecular imaging and PTT for deep tissue imaging, while nanostars can predominantly be utilized in biosensing, PTT, imaging, and SERS. The nanoshells can be helpful in biosensing, photoacoustic imaging, and PTT, where deep tissue penetration and effective thermal conversion are significant. Nanochains and nanoclusters are suitable for biosensing and molecular imaging applications, where easy clearance from the body system and detection of specific biomolecular interactions with high sensitivity are significant. Confronting regulatory approval, clinician training, and public acceptance challenges is essential for the successful implementation of GNP-based technologies on a large scale. Through extensive research, employing innovative techniques, and fostering collaboration, GNPs have the potential to enhance the capabilities of retinal imaging. Although further research and regulatory approval are needed, GNPs show significant potential for advancing retinal imaging technologies and improving outcomes. This development is expected to result in increased accuracy in diagnosis and better treatment outcomes, ultimately advancing personalized diagnostics or treatment in ophthalmology.

## Figures and Tables

**Figure 1 ijms-25-09315-f001:**
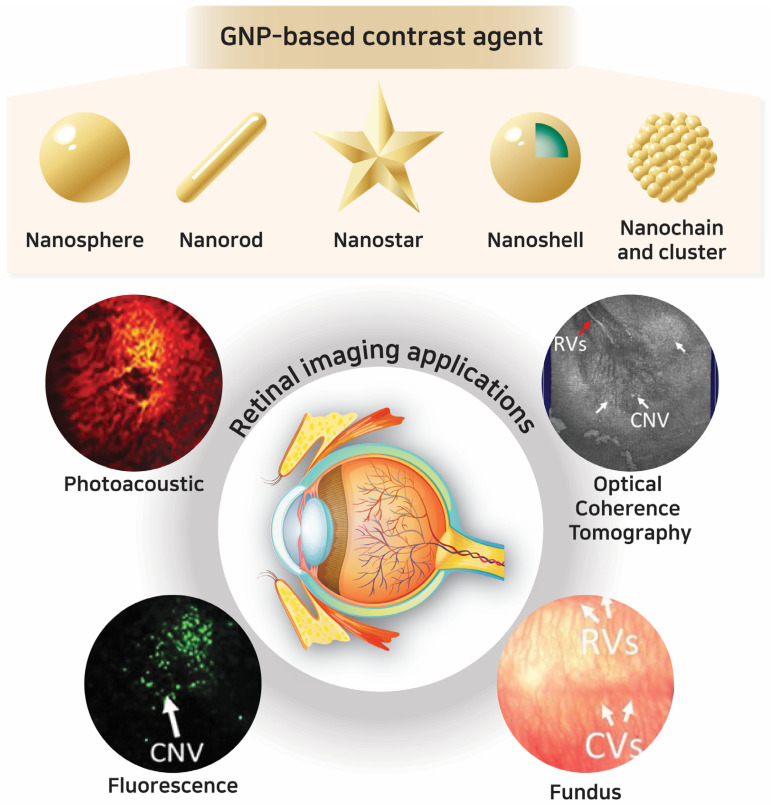
Gold nanoparticle-based contrast agents in various morphologies are introduced for retinal imaging applications.

**Figure 2 ijms-25-09315-f002:**
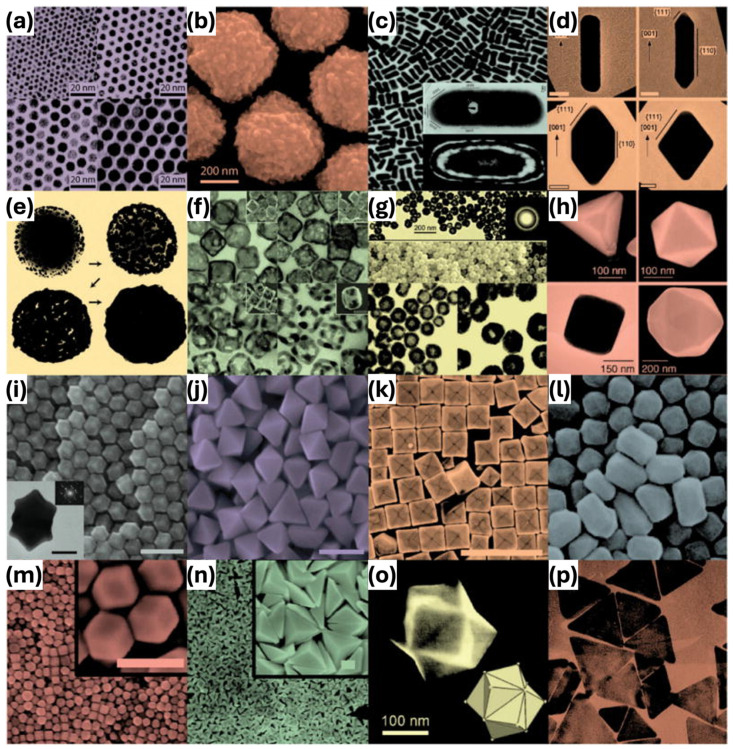
GNPs with various morphologies. (**a**) Small nanosphere, (**b**) large nanospheres, (**c**) nanorods, (**d**) sharpened nanorods, (**e**) nanoshells, (**f**) nanocages/frames, (**g**) hollow nanospheres, (**h**) tetrahedra/octahedra/cubes/icosahedra, (**i**) rhombic dodecahedra, (**j**) octahedra, (**k**) concave nanocubes, (**l**) tetrahexahedra, (**m**) rhombic dodecahedra, (**n**) obtuse triangular bipyramids, (**o**) trisoctahedra, and (**p**) nanoprisms. Used with permission from [35]; permission conveyed through Copyright Clearance Center, Inc., Danvers, MA, USA.

**Figure 3 ijms-25-09315-f003:**
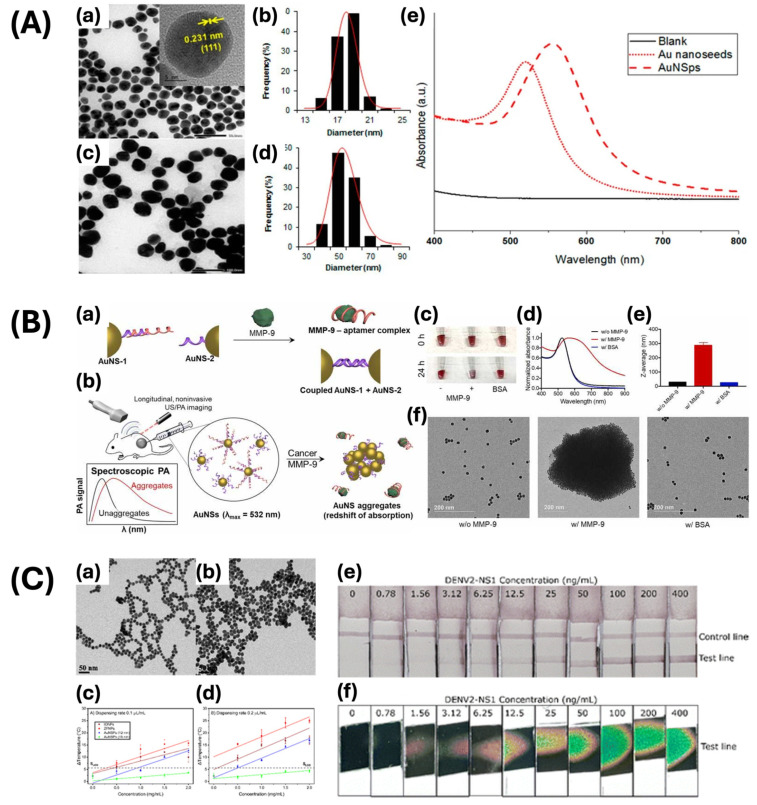
(**A**) Transmission electron microscope (TEM) images and size distribution of gold (**a**,**b**) nanoseeds and (**c**,**d**) nanospheres. The average diameters are 18.27 ± 0.08 nm and 53.52 ± 0.36 nm, respectively. The inset in (a) is a high−resolution transmission electron microscopy (HRTEM) image of gold nanoseeds. (**e**) Ultraviolet−visible (UV−Vis) spectra of gold nanoseeds and gold nanospheres show a shift towards longer wavelengths, with the peak of the SPR moving from 521 nm to 560 nm. Reproduced from [47]. Licensed under CC BY 4.0. (**B**) (**a**) The design of the plasmon coupling−based matrix metalloproteinase (MMP) sensor involves a DNA displacement mechanism induced by the presence of human matrix metalloproteinase−9 (hMMP−9). (**b**) The detection of MMP−9 in the tumor microenvironment is achieved using ultrasound−guided spectroscopic photoacoustic imaging. (**c**) The visual changes in the MMP sensor are shown under different conditions: without hMMP−9, with hMMP−9, and with bovine serum albumin (BSA). These changes are observed both immediately (0 h, upper image) and after 24 h (lower image). (**d**) The UV−Vis spectra of the MMP sensor are depicted for the conditions without hMMP−9, with hMMP−9, and with BSA after 24 h. (**e**) The hydrodynamic size measurements of the MMP sensor are presented for the conditions without hMMP−9, with hMMP−9, and with BSA after 24 h. (**f**) TEM images illustrate the morphology of the MMP sensor under different conditions: without hMMP−9 (left), with hMMP−9 (middle), and with BSA (right). The scale bar represents 200 nm. Used with permission from [48]; permission conveyed through Copyright Clearance Center, Inc., Danvers, MA, USA. (**C**) TEM images of gold nanospheres (AuNPs) with (**a**) 5 mL of trisodium citrate having an average diameter of 12.03 ± 1.18 nm, and (**b**) 2.5 mL of trisodium citrate with an average diameter of 17.96 ± 2.59 nm. The relationship between the temperature gradients and mass concentrations of AuNSPs with (**c**) 5 mL of trisodium citrate and (**d**) 2.5 mL of trisodium citrate. (**e**) The lateral flow assays (LFAs) and (**f**) thermochromic sheets of the colorimetric thermal sensing LFA strips employing 12 nm diameter AuNSPs against DENV2-NS1 show the differences at various concentrations ranging from 0 to 400 ng mL^−1^. Reproduced from [52]. Licensed under CC BY 3.0.

**Figure 4 ijms-25-09315-f004:**
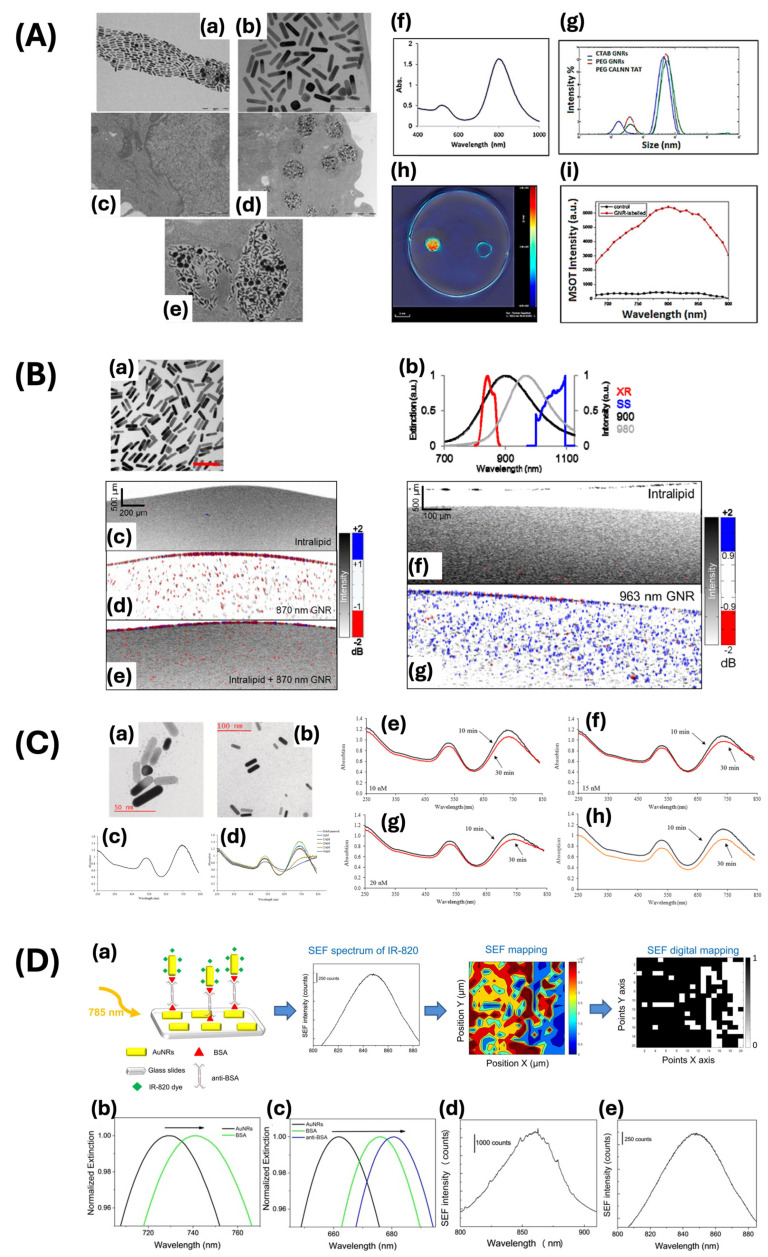
(**A**) TEM images of PEG−Cys−Ala−Leu−Asn−Asn (CALNN)−TAT gold nanorods (GNRs) show (**a**) GNRs with a 200 nm scale bar, (**b**) GNRs with a 100 nm scale bar, (**c**) stem cell control (without GNRs), (**d**) stem cells incubated with 0.5 nM PEG−CALNN−TAT GNRs for 24 h with a 1000 nm scale bar, and (**e**) stem cells with GNR−loaded endosomes and a 200 nm scale bar. (**f**) UV−Vis spectrum of functionalized GNRs demonstrates a longitudinal surface plasmon peak at around 800 nm, (**g**) shows dynamic light scattering (DLS) size distribution of GNRs with different surface coating, (**h**) represents photoacoustic imaging signal intensity at 795 nm overlapping a photograph of the phantom (left: cells loaded with GNRs, right: control cells), (**i**) indicates photoacoustic imaging signal intensity varying at different wavelengths. Reproduced with permission from [53]. (**B**) (**a**) TEM image showing PEGylated gold nanorods (GNRs) functionalized with Tat peptides for cellular uptake. The bar is 100 nm. (**b**) Normalized extinction spectra for GNRs with SPR peaks at 900 nm (10 × 50 nm GNRs, black line) and 980 nm (10 × 59 nm GNRs, gray line) compared to OCT profiles (commercial OCT, red line and swept−source OCT, blue line). (**c**) OCT images of 0.1% intralipid, (**d**) dilute GNR with SPR at 870 nm, and (**e**) intralipid mixed with GNRs, demonstrating the contrast enhancement due to GNRs with SPR peaks at 870 nm and 963 nm against an intralipid background. (**f**) OCT images of 0.1% intralipid show sparse color pixels due to noise, while (**g**) dilute GNR with SPR at 963 nm represents the GNR signal in blue. Reprinted/adapted with permission from [54]. Copyright Optical Society of America. (**C**) TEM images of gold nanorods having an average aspect ratio of approximately 4.9, with dimensions measuring 24 ± 1 nm in diameter and 118 ± 1 nm in length with (**a**) 50 nm scale bar and (**b**) 100 nm scale bar. UV−Vis absorption spectrum of (**c**) the diluted gold nanorods shows distinct plasmon resonance bands at 537 nm (transverse band) and 740 nm (longitudinal band), respectively, and (**d**) different concentrations of aptamer with gold nanorod solution, demonstrating a decrease in longitudinal plasmon band absorption. Biosensor responses to different concentrations of target CRP protein of (**e**) 10 nM, (**f**) 15 nM, and (**g**) 20 nM. (**h**) Response of biosensor to non−target protein demonstrating that only the dispersity of gold nanorods was impacted with time and that there is no evidence in UV−vis spectrum. Used with permission from [76]; permission conveyed through Copyright Clearance Center, Inc., Danvers, MA, USA. (**D**) (**a**) Schematic illustration shows the detection principle of a gold nanorod−based surface−enhanced fluorescence (SEF) biosensor. Extinction spectra for gold nanorod surface modification with (**b**) IR−820 dye (the reporter molecule) and BSA (IR−S−AuNRs), and (**c**) adsorption of AuNRs on glass slides (AuNRs−chip). Both indicate the plasmonic band redshifts 12 nm for IR−S−AuNRs and 16 nm for AuNRs-chip. SEF spectrum of (**d**) IR−820 dye (1.0 × 10^−5^ mol L^−1^) after BSA immobilization on IR−S−AuNRs and (**e**) AuNRs−chip immunoassay after detection of anti−BSA with excitation at 785 nm. Reprinted from [77]. Copyright 2023, with permission from Elsevier.

**Figure 5 ijms-25-09315-f005:**
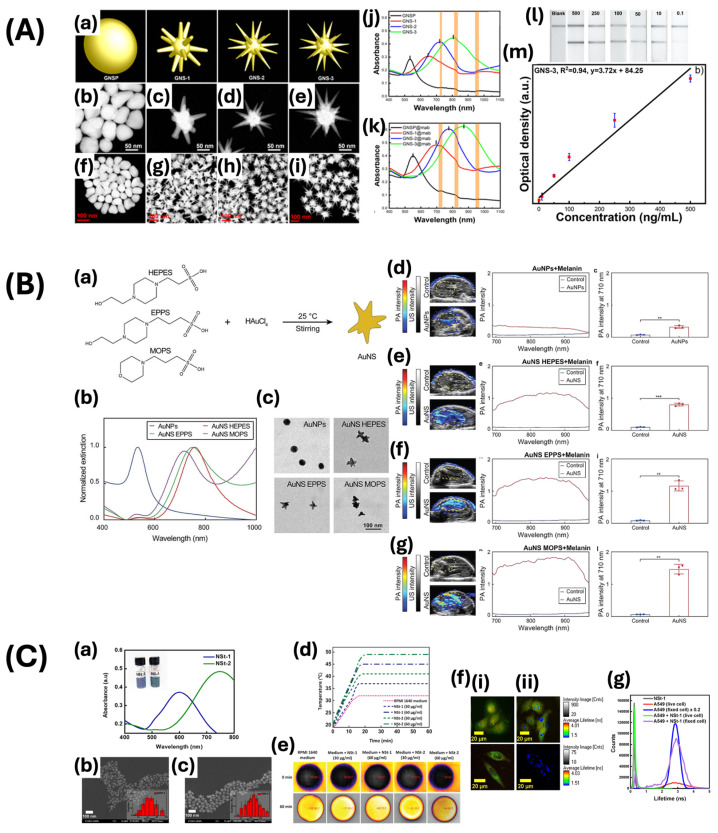
(**A**) (**a**) Three-dimensional model and scanning transmission electron microscopy (STEM) images of (**b**) gold nanosphere with 50 nm diameter spherical core and gold nanostars with different spike lengths of (**c**) 70 ± 10 nm, (**d**) 80 ± 7, and (**e**) 95 ± 7 nm based on (**b**). (**f**–**j**) STEM images of synthesized nanoparticles demonstrating high reproducibility. UV-vis absorbance spectra of (**j**) GNSPs and GNSs after synthesis, and (**k**) after antibody conjugation. (**l**) shows lateral flow immunoassay (LFIA) results and (**m**) a calibration curve for gold nanostars with 95 ± 7 nm spike length. Reprinted (adapted) with permission from [60]. Copyright 2023 American Chemical Society. (**B**) (**a**) The synthesis process of gold nanostars (AuNS). (**b**) Extinction spectra of gold nanostars. (**c**) TEM images of gold nanoparticles and gold nanostars. Photoacoustic–ultrasound (PA-US) imaging of melanin-coated (**d**) gold nanoparticles, (**e**) AuNS HEPES, (**f**) AuNS EPPS, and (**g**) AuNS MOPS in gelatin phantoms at 710 nm. Figures are modified from the original, created by [63], published in the Journal of Nanobiotechnology, 2024. Licensed under CC BY 4.0. *** is *p* < 0.001, ** is *p* < 0.01. (**C**) (**a**) UV-vis absorption spectra of gold nanostars (AuNSts) with different sizes of nanoseeds (3 nm—AuNSt-1, 15 nm—AuNSt-2) and SEM images of AuNSts and size distribution of (**b**) AuNSt-1 and (**c**) AuNSt-2. (**d**) The temperature elevation results under NIR LED light irradiation at 730 nm, and (**e**) corresponding thermal photographs of samples with different concentrations (30 and 60 μg/mL). (**f**) Fluorescent lifetime images of (**i**) control and (**ii**) A549 cells treated with AuNSts, excited at 780 nm wavelengths with 14 mW power for (**i**) and 5 mW for (**ii**). (**g**) Histogram of lifetime distribution along the images. Reproduced (reused) with permission from [65]. Copyright 2021 John Wiley & Sons, Inc., Hoboken, NJ, USA.

**Figure 6 ijms-25-09315-f006:**
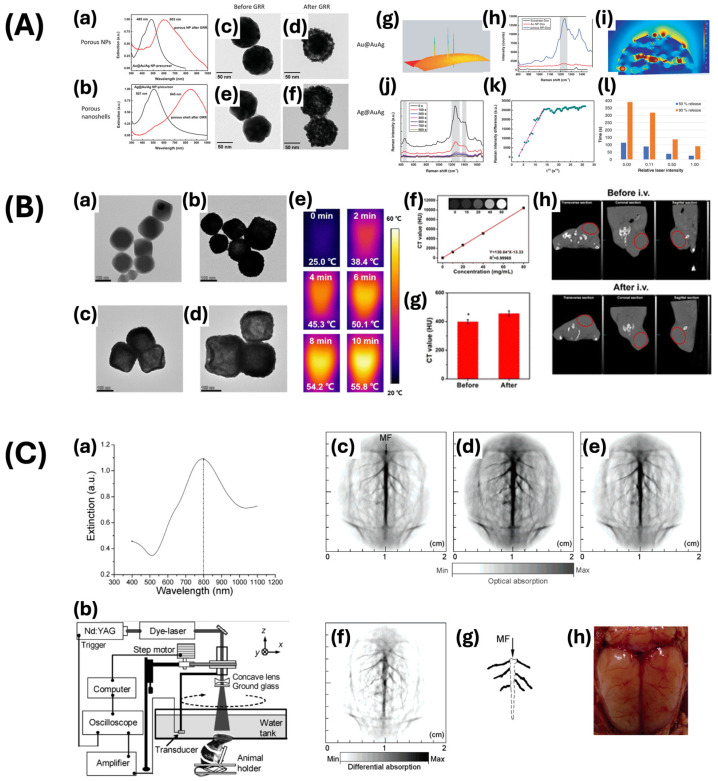
(**A**) Extinction spectra of (**a**) Au@Au/Ag and (**b**) Ag@Au/Ag nanoshells. TEM images of (**c**) before and (**d**) after galvanic replacement reaction (GRR) of Au@Au/Ag, and (**e**) before and (**f**) after GRR of Ag@Au/Ag nanoshells. (**g**) Three-dimensional surface plot illustrating the scattering of individual porous nanoparticles observed by back-reflection mode microscopy. (**h**) Raman signals obtained from a glass substrate, a single 100 nm gold nanoparticle coated with DOX, and 96 nm porous nanoparticles that were put on the glass substrate. (**i**) The simulated local field enhancement of porous nanoparticles under 633 nm laser irradiation. (**j**) SERS spectra obtained from DOX-loaded porous Au-Ag nanoparticles after laser exposure. (**k**) The intensity decay at the Raman peak around 1250 cm^−1^ over 1000 s in steps of 10 s with the square root of time. (**l**) The time required for 50% and 90% DOX release under different laser intensities, where a relative intensity value of 1.0 corresponds to 7.7 × 10^4^ W cm^−2^. Used with permission from [67]; permission conveyed through Copyright Clearance Center, Inc., Danvers, MA, USA. (**B**) TEM images of (**a**) UiO-66-NH_2_, (**b**) UiO-66-NH_2_@gold nano-shells (UiO-66-NH_2_@AuNSs), (**c**) hollow gold nanoshell (HAuNS), and (**d**) HAuNS@PEG-bio nanoparticles. (**e**) The thermographic images of HAuNS@PEG-bio at 80 μg mL^−1^ concentration with 1064 nm laser irradiation at a power density of 1 W cm^−2^ for 10 min. (**f**) Computed tomography (CT) value and images of different concentrations of HAuNS@PEG-bio. In vivo (**g**) CT value (* *p* < 0.05) and (**h**) CT images of tumor locations in mice before and after intravenous injection of HAuNS@PEG-bio. Used with permission from [78]; permission conveyed through Copyright Clearance Center, Inc., Danvers, MA, USA. (**C**) (**a**) Extinction spectrum of the gold–silica nanoshell with the peak absorbance at 800 nm. (**b**) Photoacoustic tomography (PAT) system diagram for in vivo rat brain imaging. Photoacoustic image obtained (**c**) before, and (**d**) 20 min and (**e**) 350 min after introduction of nanoshells. (**f**) Differential image acquired by subtracting the pre-injection image (**c**) from the post-injection image (**d**). (**g**) The segmented large blood vessels in the cerebral cortex from the PAT image. (**h**) The skull-removed image of the rat brain cortex taken after PAT data acquisition. Reprinted (adapted) with permission from [84]. Copyright 2004 American Chemical Society.

**Figure 7 ijms-25-09315-f007:**
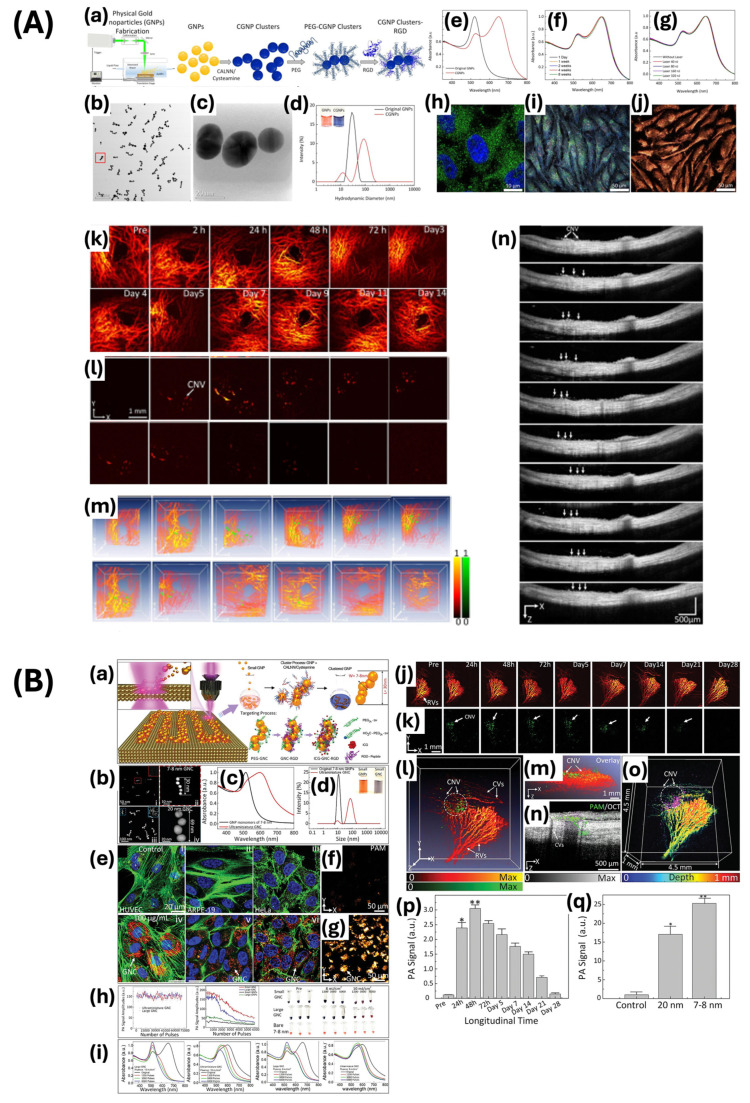
(**A**) (**a**) Diagram illustrating the CGNP cluster–RGD synthesis process. (**b**) TEM image of CGNP cluster–RGD with a magnification of 2000× and accelerating voltage of 20 kV. The scale bar represents a length of 200 nm. (**c**) A high magnification view of CGNP cluster–RGD, marked by the red square in (**b**), showing the CALNN-PEG-RGD layer within a region within CGNP cluster–RGD. (**d**) Dynamic light scattering results indicating the hydrodynamic diameters of GNPs and CGNP cluster–RGD. UV-Vis absorption spectra of (**e**) GNPs and CGNP cluster–RGD, and (**f**) CGNP cluster–RGD colloidal stability over 8 weeks for quantitative evaluation. (**g**) Photostability assessment of in vitro CGNP cluster–RGD under nanosecond pulsed laser irradiation (65 s) at different pulse fluences (0.005, 0.01, 0.02, and 0.04 mJ/cm^2^). (**h**) Confocal fluorescence microscopy image of single HeLa cell treated with FITC-labeled CGNP cluster–RGD for 24 h at a concentration of 50 µg/mL. Hoechst 33342-stained cell nuclei are indicated by the blue fluorescence, and the CGNP cluster–RGD that have accumulated surrounding the nuclei are visible with the green fluorescence. HeLa cells co-cultured with (**i**) 20 nm diameter of conventional GNP-RGD and (**j**) CGNP cluster–RGD in dark field image, displaying the distribution throughout the cells. The PAM images of CNV acquired at (**k**) 578 nm and (**l**) 650 nm wavelengths before and after injecting 0.5 mL of CGNP cluster–RGD at 2.5 mg/mL. (**m**) Three-dimensional visualization of the data displayed in (**k**,**l**). The red color represents the retinal vasculature, whereas the green represents the distribution of CGNP cluster–RGD. (**n**) B-scan OCT images at different time points after injecting CGNP cluster–RGD (2 h, 24 h, 48 h, 73 h, 4, 5, 7, 9, 11, and 14 days). Reprinted (adapted) from [7]. Licensed under CC BY 4.0. (**B**) (**a**) Diagram illustrating the procedure for preparing GNC-RGD with RGD ligands and indocyanine green (ICG) coating. (**b**) TEM images of ultraminiature GNC-RGD composed of (**i**) 7–8 nm GNP spheres, (**ii**) higher magnification of (**i**), (**iii**) 20 nm GNP spheres, and (**iv**) higher magnification of (**iii**). (**c**) UV-vis absorption spectra of 7–8 nm GNP and ultraminiature GNC. (**d**) Size distribution of GNP and GNC measured by the dynamic light scattering. (**e**) Confocal microscopy images of HUVEC, ARPE-19, and HeLa cells treated with 100 μg mL^−1^ of (**i**–**iii**) non-targeting GNC, and (**iv**–**vi**) GNC-RGD for 24 h. PAM images of ARPE-19 cells treated with (**f**) non-targeting GNS, and (**g**) with targeting GNCs. (**h**) PA signals produced by ultraminiature GNC having 7–8 nm diameter and larger GNCs having 20 nm diameter. In addition, the threshold damage of those GNCs, and photographs of samples before and after laser irradiation. (**i**) UV-vis absorption spectra of GNCs after laser irradiation at different exposure pulses of 1200, 3000, and 6000 pulses and laser fluences of 8 and 10 mJ cm^−2^. Maximum intensity projection (MIP) PAM images using (**j**) 578 nm wavelength displaying the structure of retinal vessels, choroidal vessels, and capillaries, and (**k**) 650 nm wavelengths showing CNV. (**l**) Selected integrated 3D representation of CNV at 48 h. (**m**) XZ orientation of the overlaid PAM image. (**n**) OCT image and PAM image acquired at a wavelength of 650 nm. (**o**) PAM image with depth information. (**p**) The time-dependent amplitude of the PAM signal from the region of interest in (**k**). (**q**) Comparison of PAM signal amplitude among control (non-targeted ultraminiature GNC), targeting ultraminiature GNC, and targeting large GNC, presented as mean ± standard deviation (N = 3, * is for *p* < 0.05, ** is for *p* < 0.01). Reprinted (adapted) with permission from [8]. Licensed under CC BY 4.0.

**Figure 8 ijms-25-09315-f008:**
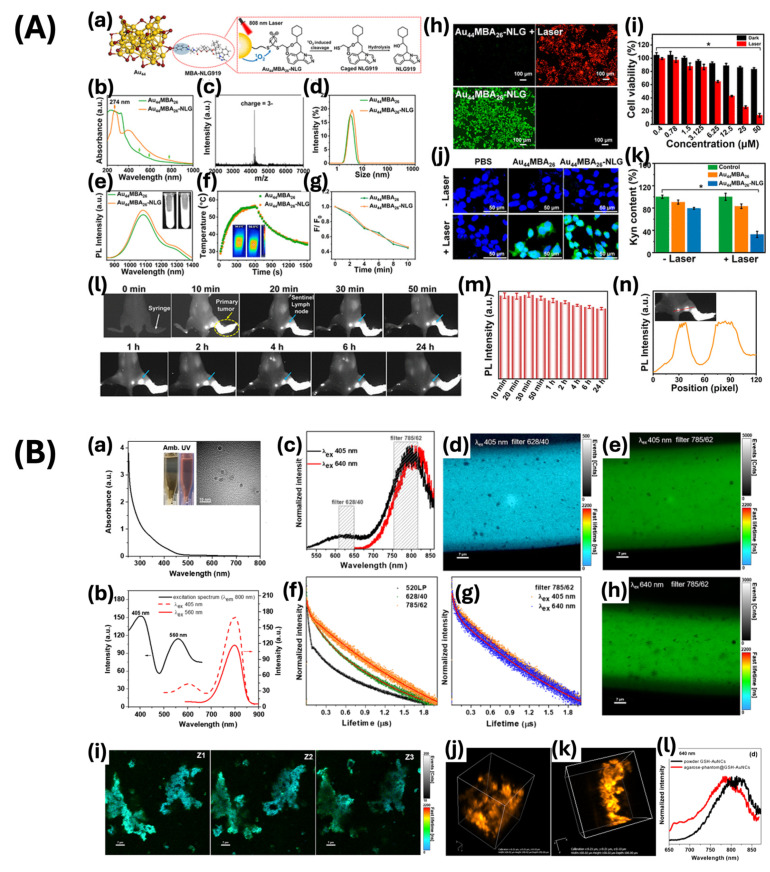
(**A**) (**a**) Diagram of Au_44_MBA_26_−NLG NIR photoactivation mechanism. (**b**) UV−Vis absorption spectra of Au_44_MBA_26_ and Au_44_MBA_26_−NLG. (**c**) The electrospray ionization–mass spectrometry of Au_44_MBA_26_. (**d**) Dynamic light scattering analysis of Au_44_MBA_26_ and Au_44_MBA_26_−NLG. (**e**) Photoluminescence emission spectra at 808 nm excitation of Au_44_MBA_26_ and Au_44_MBA_26_−NLG, including the photos of the sample. (**f**) Photothermal temperature change in Au_44_MBA_26_ and Au_44_MBA_26_−NLG with 808 nm laser irradiation at 1.5 W cm^−2^, including infrared thermal images. (**g**) The productivity of singlet oxygen (^1^O_2_) relative to Au_44_MBA_26_ and Au_44_MBA_26_−NLG varies with the duration of laser irradiation. (**h**) Photoluminescence images of 4T1 cells treated with Au_44_MBA_26_−NLG and exposed to 808 nm laser irradiation for 6 min at an intensity of 1.5 W cm^−2^ (top) or kept in dark conditions (bottom). Viable and nonviable cells are stained green and red by Calceim AM and EthD−1, respectively. (**i**) Cytotoxicity and photothermal effects of Au_44_MBA_26_−NLG on 4T1 cells with different concentrations in the absence of light and under laser irradiation, presented as mean ± standard deviation (N = 5, * *p* < 0.05). (**j**) Confocal photoluminescence images of 4T1 cells treated with phosphate-buffered saline (PBS) as a control, or Au_44_MBA_26_ and Au_44_MBA_26_−NLG nanoparticles for a duration of 12 h. The cells were then stained with H_2_DCFDA and exposed to laser irradiation for 6 min. The emission of a green photoluminescent signal is observed from the oxidized form of 2′,7′−dichlorofluorescein (DCF) after interaction with ^1^O_2_. (**k**) Relative kynurenine (Kyn) content in the culture medium of 4T1 cells treated with control, Au_44_MBA_26_, or Au_44_MBA_26_−NLG for 24 h, with or without 6 min of laser irradiation, presented as mean ± standard deviation (N = 3, * is for *p* < 0.05). (**l**) NIR II photoluminescence images of the 4T1 tumor−bearing mice taken at various intervals following Au_44_MBA_26_−NLG intratumoral injection. (**m**) Temporal variations in the relative photoluminescence intensities of Au_44_MBA_26_−NLG at the tumor location. (**n**) The profile of photoluminescence intensity obtained in the region of interest by following the white lines. Reprinted (adapted) with permission from [71]. Copyright 2023 American Chemical Society. (**B**) (**a**) UV−Vis absorption spectrum of GSH−AuNCs with images of colloidal aqueous suspension under normal light and UV light, along with a representative HR−TEM. (**b**) The excitation spectrum at 800 nm wavelength and photoluminescence spectrum at 405 and 560 nm wavelength of GSH−AuNCs. (**c**) Photoluminescence spectra of AuNCs in powder form excited at 405 and 640 nm wavelength. The representative fluorescence lifetime imaging microscopy images under 405 nm excitation with (**d**) 628/40 and (**e**) 785/62 emission filter. (**f**) Fluorescence lifetime decay curves under 405 nm excitation using 520LP, 628/40, and 785/62 emission filters. (**g**) Fluorescence lifetime decay curves under 405 nm and 640 nm excitation using a 785/62 emission filter. (**h**) The representative fluorescence lifetime imaging microscopy images under 640 nm excitation with 785/62 emission filter. (**i**) The representative fluorescence lifetime imaging microscopy images of GSH−AuNCs embedded in an agarose phantom at different z−depths with an excitation wavelength of 640 nm and a 647LP emission filter. (**j**,**k**) The representative 3D re-scan confocal fluorescence imaging of the agarose phantom with GSH−AuNCs, excited at a wavelength of 640 nm. (**l**) Photoluminescence spectra of the powder form and agarose phantom with GSH−AuNCs. Used with permission from [72]; permission conveyed through Copyright Clearance Center, Inc., Danvers, MA, USA.

**Figure 9 ijms-25-09315-f009:**
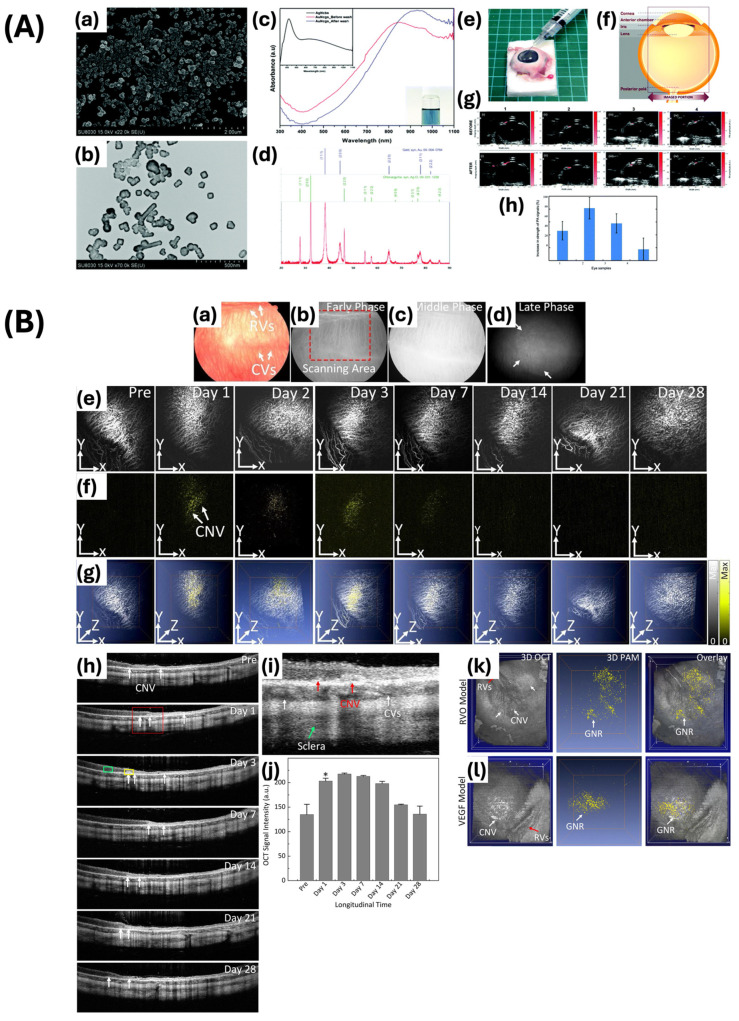
(**A**) (**a**) Scanning electron microscopy (SEM) image and (**b**) inverted-alpha SEM image of AuNcgs. (**c**) UV-Vis absorption spectra of AgNcbs (inset) and AuNcgs-before (red) and after (blue) wash. (**d**) X-ray powder diffraction spectra of AuNcgs with standard peaks of Au (00-004-0784) and AgCl (00-031-1238) taken from JCPDS database. (**e**) Injection of AuNcgs into the porcine eye sample. (**f**) Schematic diagram of the eye. (**g**) The integrated images of eye samples 1 to 4, displaying the effects before and after injecting AuNcgs. The post-injection image demonstrates a noticeable rise in photoacoustic signal intensity. (**h**) Increase in photoacoustic signal strength after injecting AuNcgs. Eye samples 1 to 4 showed increases of 46.3%, 81.4%, 57.9%, and 17.6%, respectively. Used with permission from [94]; permission conveyed through Copyright Clearance Center, Inc., Danvers, MA, USA. (**B**) (**a**) Color fundus photography of the CNV rabbit model. Indocyanine green angiography (ICGA) images of (**b**) early, (**c**) middle, and (**d**) late phase, with white arrows indicating the growth of CNV. The acquired XY MIP images at (**e**) 578 nm and (**f**) 700 nm (pseudo-yellow color shows the dispersion of GNRs surrounding the CNV) before and after injection at days 1, 3, 7, 14, 21, and 28. (**g**) The overlaid 3D PAM images of (**e**,**f**). (**h**) Long-term B-scan OCT images showing the location of CNV before and on days 1, 3, 7, 14, 21, and 28 post-intravenous injection of GNRs, with white arrows indicating CNV and yellow and green rectangles highlighting region of interest for OCT signal intensity analysis. (**i**) Magnified view of the selected region in (**i**), illustrating the surface margin of CNV in red arrows and different layers such as choroidal vessels and the sclera. (**j**) OCT signal intensity over time, demonstrating a rise that peaked on day three following injection at 161.3%. * is for *p* < 0.05. Fusion *en-face* volumetric PAM and OCT images of (**k**) retinal vein occlusion (RVO)-induced CNV model, with red arrows indicating the morphology of retinal vessels, and (**l**) subretinal injection of VEGF model, with white arrows indicating the developed CNV with high contrast on 3D OCT images. Reprinted (adapted) with permission from [95]. Copyright 2021 American Chemical Society.

**Figure 10 ijms-25-09315-f010:**
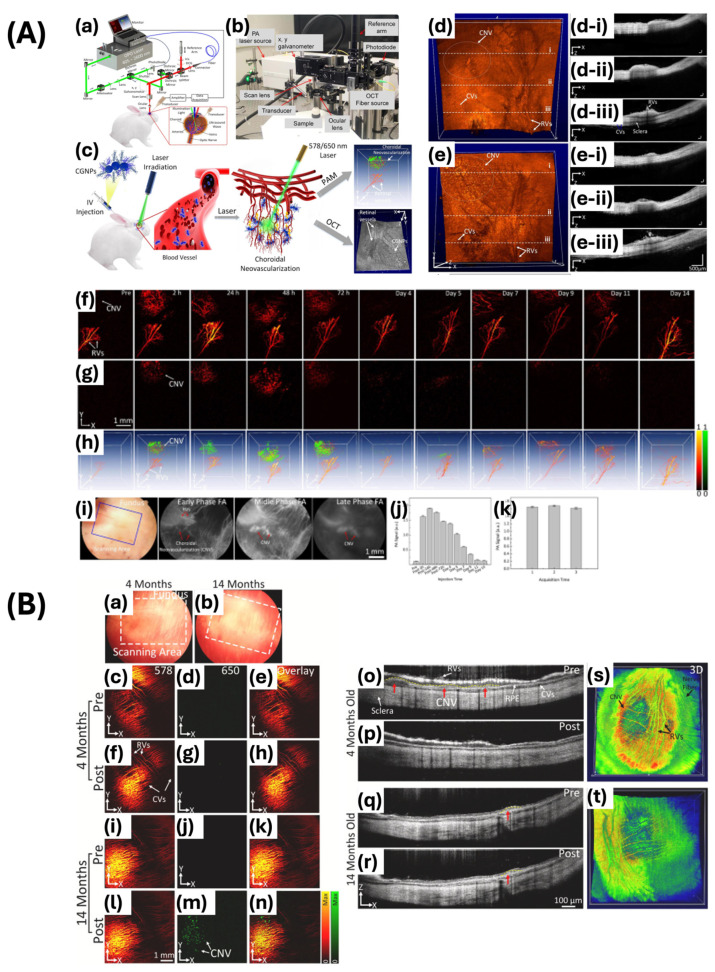
(**A**) (**a**) Schematic diagram and (**b**) physical setup of the imaging system used for multimodal imaging of the retina. (**c**) demonstrates the utilization of intravenously injected CGNP cluster–RGD for multimodal imaging in a rabbit model. The imaging technique involves generating photoacoustic signals from the retina by irradiating nanosecond pulsed laser at 578 or 650 nm wavelengths. Three-dimensional OCT images of (**d**) pre-injection of CGNP cluster–RGD, with (**di**–**diii**) showing B-scan OCT images of a vertical slice (white dotted lines) through the retina in (**d**), and (**e**) post-injection of CGNP cluster–RGD, with (**ei**–**eiii**) displaying B-scan OCT images of a vertical slice (white dotted lines) through the retina in (**e**). PAM image of CNV before and after injection of 0.5 mL CGNP cluster–RGD at a concentration of 2.5 mg/mL, acquired using (**f**) 578 nm and (**g**) 650 nm wavelengths. (**h**) Overlay 3D images showing the distribution of CGNP cluster–RGD accumulated at the CNV. (**i**) Color fundus and fluorescein angiography (early, middle, and late phases) images of the retina. (**j**) Photoacoustic signal in the rabbit post-injection of CGNP cluster–RGD, exhibiting a considerable increase in signal intensity after injection, reaching its peak at 24 h and subsequently decreasing progressively. (**k**) In vivo photostability of CGNP cluster–RGD with error bars indicating the standard error of the average photoacoustic signal. Reprinted (adapted) with permission from [7]. Licensed under CC BY 4.0. (**B**) Color fundus photographs of rabbits at (**a**) 4 months of age, and (**b**) 14 months of age, taken 28 days after therapy. PAM images at (**c**) 578 nm and (**d**) 650 nm before CGNP delivery (400 μg/mL) in 4-month age group, with (**e**) the overlaid images of (**c**,**d**). PAM images at (**f**) 578 nm and (**g**) 650 nm after CGNP delivery in 4-month age group, with (**h**) the overlaid images of (**f**,**g**). PAM images at (**i**) 578 nm and (**j**) 650 nm before CGNP delivery in 14-month age group, with (**k**) the overlaid images of (**i**,**j**). PAM images at (**l**) 578 nm and (**m**) 650 nm after CGNP delivery in 14-month age group, with (**n**) the overlaid images of (**l**,**m**). OCT image of the 4-month age group (**o**) before and (**p**) after treatment. OCT images of the 14-month of age group (**q**) before and (**r**) after treatment. Three-dimensional OCT images from the (**s**) 4-month and (**t**) 14-month age groups. Used with permission from [96]; permission conveyed through Copyright Clearance Center, Inc., Danvers, MA, USA.

**Figure 11 ijms-25-09315-f011:**
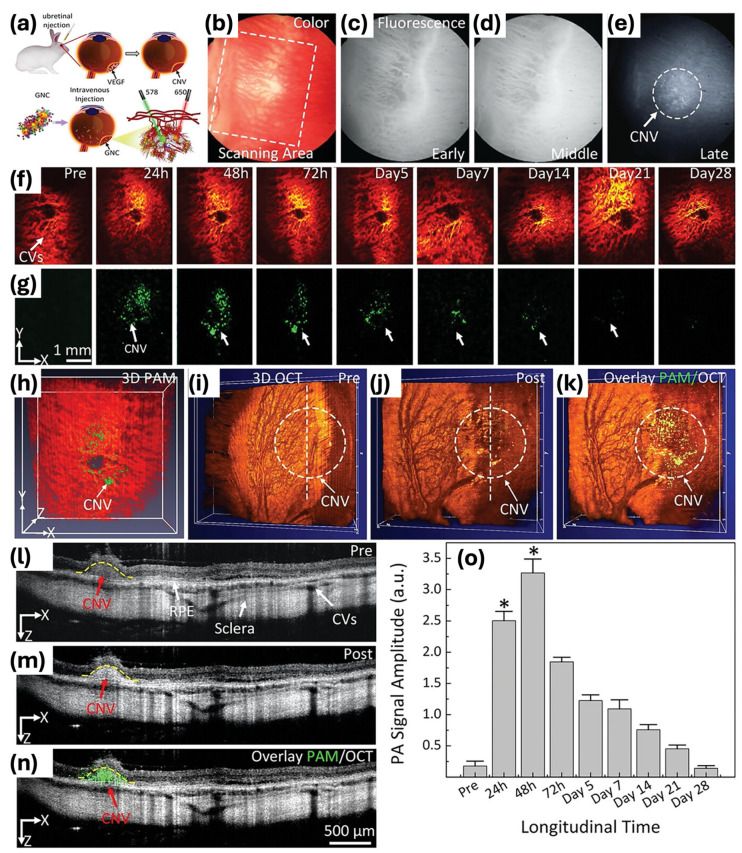
(**a**) Diagram illustrating the process of injecting VEGF into the subretinal region, followed by intravenous administration of targeting GNCs. (**b**) Color fundus photograph of the retina. Indocyanine green angiography (ICGA) images acquired at (**c**) early, (**d**) middle, and (**e**) late phases prior to the injection of ultraminiature GNCs. The acquired PAM images at (**f**) 578 nm and (**g**) 650 nm. The areas of CNV are indicated by white arrows. (**h**) The overlaid 3D PAM images at 578 and 650 nm. Three-dimensional OCT images were taken (**i**) before and (**j**) following treatment, with the location of CNV being highlighted by a white dotted circle. (**k**) The overlayed PAM and OCT images. Two-dimensional OCT images (**l**) before and (**m**) after treatment depicting the retinal layers, including RPE, choroid, choroidal vessels, CNV, and sclera. (**n**) The correlated 2D PAM and OCT images, with green showing the presence of ultraminiature GNCs in the CNV region. (**o**) Quantitative measurements of PA signals in CNV at different time points, presented as mean ± standard deviation (N = 3, * *p* < 0.05). Reprinted (adapted) from [8]. Licensed under CC BY 4.0.

**Figure 12 ijms-25-09315-f012:**
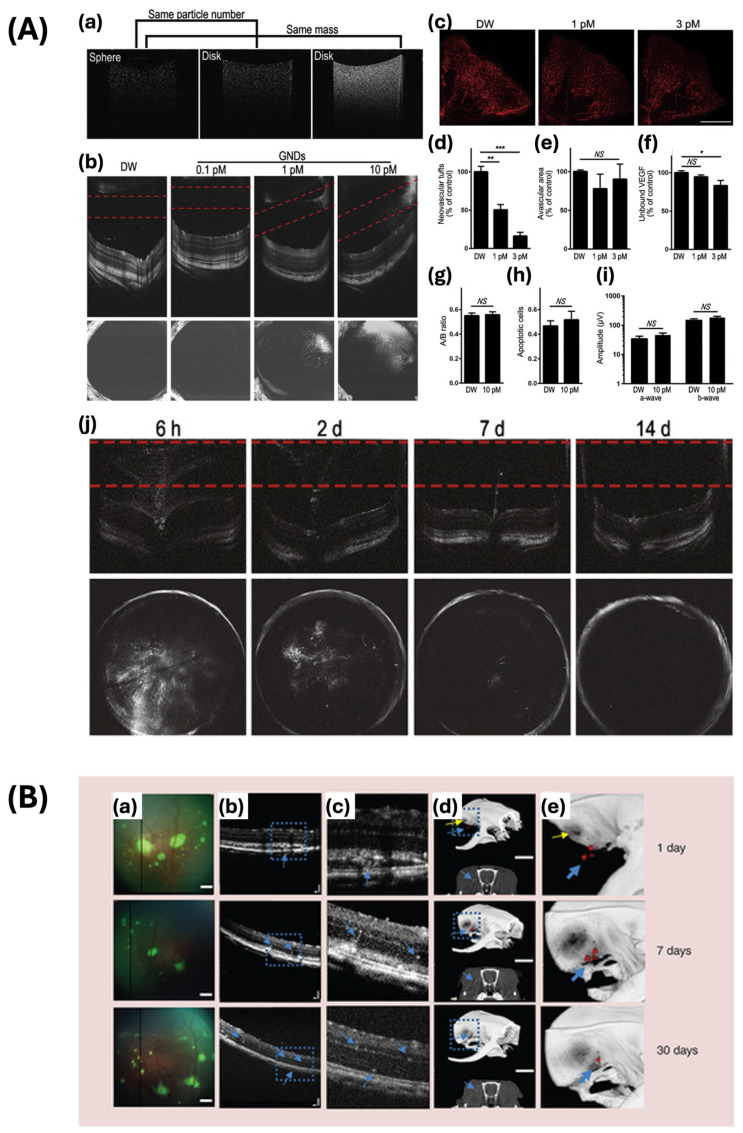
(**A**) (**a**) Cross-sectional OCT images of different solutions. The left image is of a 1 pM solution containing 200 nm gold nanospheres, the center image is of a 1 pM solution containing 160 nm GNDs, and the right image is of a 10.4 pM solution containing 160 nm GNDs. Those have the same mass concentration as the 1 pM 200 nm gold nanospheres. (**b**) Cross-sectional OCT images (upper) of a mouse eye taken immediately after injecting distilled water and 160 nm GNDs at concentrations of 0.1, 1, and 10 pM, along with projected images (lower) from the region between the red dashed lines. (**c**) Immunostaining of isolated retinas was performed using the anti-isolectin-B4 antibody at P17, after intravitreal injection of 0.5 μL DW and GND solutions at concentrations of 1 pM and 3 pM at P14. Quantitative assessment of the anti-angiogenic action of GNDs by measuring (**d**) neovascular tufts and (**e**) avascular area. *** is *p* < 0.001, ** is *p* < 0.01, and * is *p* < 0.05. NS denotes not significant. (**f**) The levels of unbound VEGF assessment in the vitreous and retina at P15 after injecting GND solutions into the vitreous at P14. (**g**) Histological analysis of the integrity of the retina 5 weeks after injecting 10 pM of GNDs. (**h**) TUNEL assay for detecting apoptotic cells in retinal slices after GND injection. (**i**) Electroretinogram analysis after injection to quantify the amplitudes of the a- and b-wave, presented as mean ± standard deviation (N = 6, *** *p* < 0.001, ** *p* < 0.01). (**j**) Cross-sectional (upper) and projectional (lower) OCT images of mouse eyes at 6 h, 2 d, 7 d, and 14 d after intravitreal injection of 160 nm GNDs at a concentration of 10 pM (N = 5). Reprinted from [97]. Copyright 2017 with permission from Elsevier. (**B**) (**a**) Fluorescence fundus camera captured an image of labeled cells with green fluorescent protein and GNPs at 1, 7, and 30 d after transplantation, with a scale bar of 0.5 mm. (**b**) OCT images of the retinal layers with transplanted cells being highlighted by arrow and a dotted-line square, with the scale bar of 500 um. (**c**) Enlarged images of the marked region in (**b**). (**d**) The 2D and 3D images of the reconstructed CT images, pointing to the location of transplanted cells in yellow arrows with a scale bar of 40 mm. (**e**) Enlarged view of the delineated region in (**d**). Used with permission from [98]; permission conveyed through Copyright Clearance Center, Inc., Danvers, MA, USA.

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
