# Peer review of "Gold Nanoparticles for Retinal Molecular Optical Imaging"

_ijms, 2024, doi:10.3390/ijms25179315_

Round 1

Reviewer 1 Report

Comments and Suggestions for Authors

The paper is intriguing, and it's clear that the authors have invested considerable effort in researching the literature on gold nanoparticles and compiling it into a comprehensive review. Their dedication and hard work are commendable. However, as a reviewer, I must offer some constructive criticism:

1)          The introduction needs to be condensed and more tightly focused on the paper's main topic.

2)      Section 2 (lines 89 to 107) largely repeats the introduction. This redundancy should be addressed by either omitting this section or merging it with the introduction to create a more concise opening.

3)       The authors dedicate an excessive amount of space to describing various forms of gold nanoparticles and their synthesis, which seems tangential to the paper's main theme. The chapter on different forms of nanogold could be substantially shortened, which would enhance the overall structure of the work. Notably, "Principles and Classification of GNPs" spans 16 pages of text and figures, merely introducing the topic while constituting over 30% of the paper's volume.

4)      The relevance of the chapter "Clinical translation and regulatory considerations" is questionable. It covers information that is generally understood, as all substances intended for medical use undergo such processes. This chapter could be either removed or significantly condensed without compromising the paper's quality.

5)      The summary lacks a definitive conclusion or suggestion regarding which of the discussed gold nanoparticles are most suitable or promising for retinal imaging.

Addressing these points would significantly improve the paper's focus, concision, and overall impact.

Reviewer 2 Report

Comments and Suggestions for Authors

General comment:

This work aims at reviewing the use of gold nanoparticles for retinal molecular optical imaging. The review copes with the synthesis, characterization, imaging methodologies and biological tests. 

Specific comments throughout the paper: 

Abstract is relatively fine, but not so catchy. 

1.

The introduction is relatively short, quite coincise. In my opinion, it can be more focused and it can be clearer to the readership if a figure summarizing the general points could be provided.

2. 

Lines 110-117: missing ref.

Line 138: missing ref.

In the synthesis sect. 2.1, please provide some quantitative details about the outputs of a given methodology (e.g., sizes, dispersion).

Lines 160-163: missing ref

Table 1 is incomplete. Several comparison points and KPIs are missing. It is not so meaningful for a review work. I suggest to improve it. 

For instance, given Sect. 2.2, several discussion and results are presented about emissivity, absorption, etc. Why the author are not using these data to populate the table and increase the quality and value of their work, as well as the analysis, for the community?

3. 

An introductory sentence or brief discussion is missing.

The quality of most of the images is not of enough quality. 

The discussion, by the way, is relatively smooth and interesting.

4. 

The challenges and regulators issues presented in this section are very appealing for the community. 

Lines 1176: the information about the photothermal therapy are relatively limited. More details can be added. 

5. 

The conclusion section is fine. 

Comments on the Quality of English Language

Minor english adjustments have to be made 

Round 2

Reviewer 1 Report

Comments and Suggestions for Authors

Thank you for dedicating a significant amount of effort to improving your article. I appreciate your thoughtful consideration of my comments. In the areas where we had differing views, you presented logical and compelling arguments in defense of your perspective. I have no further comments. Well done, and best of luck with your future research